# Organic photodiodes with bias-switchable photomultiplication and photovoltaic modes

Qingxia Liu [1], Lingfeng Li[1], Jiaao Wu [1], Yang Wang [1] ✉, Liu Yuan[1], Zhi Jiang[2], Jianhua Xiao[1], Deen Gu[1], Weizhi Li[1], Huiling Tai [1] ✉ & Yadong Jiang[1]

The limited sensitivity of photovoltaic-type photodiodes makes it indispensable to use pre-amplifier circuits for effectively extracting electrical signals, especially when detecting dim light. Additionally, the photomultiplication photodiodes with light amplification function suffer from potential damages caused by high power consumption under strong light. In this work, by adopting the synergy strategy of thermal-induced interfacial structural traps and blocking layers, we develop a dual-mode visible-near infrared organic photodiode with bias-switchable photomultiplication and photovoltaic operating modes, exhibiting high specific detectivity (~$10^{12}$ Jones) and fast response speed (0.05/3.03 ms for photomultiplication-mode; 8.64/11.14 µs for photovoltaic-mode). The device also delivers disparate external quantum efficiency in two optional operating modes, showing potential in simultaneously detecting dim and strong light ranging from ~$10^{-9}$ to $10^{-1}$ W cm$^{-2}$. The general strategy and working mechanism are validated in different organic layers. This work offers an attractive option to develop bias-switchable multi-mode organic photodetectors for various application scenarios.

Photodiodes have the advantages of low dark current, fast response, small size, and easy integration, rendering them widely used in charge-coupled device/complementary metal oxide semiconductor (CCD/CMOS) arrays and other imaging systems[1–4]. To date, organic photodiodes (OPDs) are considered promising alternatives to traditional commercial inorganic photodetectors (based on Si, InGaAs, etc.) due to the additional advantages of organic photosensitive materials, such as solution processability, adjustable spectral absorption, and mechanical flexibility[5–9]. Because the rectification characteristic of OPDs makes their photoresponse under forward bias almost indistinguishable, and thus the devices usually operate under reverse bias based on photovoltaic (PV) effect, known as PV-mode[10]. Generally, the external quantum efficiency (EQE) of photodiodes without photomultiplication (PM) effect is less than 100%, which makes the signal acquisition of a single pixel quite a challenge for both inorganic and organic photodiodes when detecting dim light signals (especially at night).

The photocurrent ($I_{ph}$) can be estimated from

$$EQE = R \cdot \frac{hc}{e\lambda} = \frac{I_{ph}}{S_{pixel} \cdot P_{in}} \cdot \frac{hc}{e\lambda} \cdot 100\% (\%) \quad (1)$$

in which $R$ is responsivity, $h$ is Planck's constant, $c$ is velocity of light, $\lambda$ is wavelength of the irradiated light, $S_{pixel}$ is the area of a single pixel, and $P_{in}$ is irradiated optical power intensity[11]. Assuming that $P_{in}$ at night is $10^{-8}$–$10^{-6}$ W cm$^{-2}$ (full moon)[12–14] and $I_{ph}$ is positively correlated with $S_{pixel}$, which is smaller than 10 µm × 10 µm according to the typical pixel size of CCD/CMOS arrays reported so far[15], the maximum calculated $I_{ph}$ of a single pixel is only ≈$10^{-12}$ A, not to mention the absence of moonlight. Therefore, a pre-amplifier circuit or the increased integration time is needed for these devices in practical applications, both of which result in complex imaging system design and high cost[16,17].

[1]State Key Laboratory of Electronic Thin Films and Integrated Devices, School of Optoelectronic Science and Engineering, University of Electronic Science and Technology of China, 610054 Chengdu, China. [2]Innovative Center for Flexible Devices (iFLEX), School of Materials Science and Engineering, Nanyang Technological University, 50 Nanyang Avenue, Singapore 639798, Singapore. ✉e-mail: landlord@uestc.edu.cn; taitai1980@uestc.edu.cn

Devices with PM effect can provide EQE much greater than 100% and exhibit strong detection capability to dim light signals, avoiding the use of pre-amplifier circuits[16,18]. The inorganic PM-type devices, which take the photomultiplier tube and avalanche photodiode as typical examples, often require high applied voltage, high vacuum environment, assisted cooling, or not compatible with the planar process of CMOS or CCD image sensor[19]. In recent decades, the PM-type OPDs have been developed to break this bottleneck, which also meets the demand of miniaturization and integration for the new generation of imaging systems[20–31]. The main accepted working mechanism of PM-OPDs is charge tunneling injection from external circuit induced by trap-assisted energy band bending at the electrode/active-layer interface. An EQE of $10^3$–$10^5$% has been achieved by improving the device performance[16,32,33], which means that the device operating in PM-mode has a multiplicated photocurrent that is about $10$–$10^3$ times larger than that in PV-mode (even if the EQE in PV-mode reaches a maximum of 100%). Considering that a much higher bias usually is required for driving devices in PM-mode, it will bring a large power consumption to the imaging system[7,11]. The non-negligible power consumption brings potential damages (heat-dissipation, breakdown, etc.) to the organic active layer, especially under strong daylight (-$10^{-1}$ W cm$^{-2}$)[34]. Furthermore, suffering from the slow charge-accumulation and band-bending process, the response speed of PM-OPDs is much slower, especially the charge-releasing-determined fall time, which is quite an obstacle to the practical application[35,36].

Since the trapped and tunneling injected carrier type is difficult to be tuned by the bias direction, devices that can simultaneously implement effective bias-switchable PM/PV modes have not been reported. Herein, we present the first dual-mode vis-NIR OPD that can operate in PM/PV mode under forward/reverse bias for different application demands. The key point for achieving dual operating modes is to induce or block tunneling carriers under different bias directions. We design the OPD to operate in PM-mode under forward bias by introducing thermal-induced interfacial structural traps, which can tune the trapped carrier type by bias direction; and retain the PV-mode under reverse bias by introducing a $MoO_3$ blocking layer to prevent tunneling electron injection. The structural traps at ZnO/bulk-heterojunction (BHJ) and BHJ/$MoO_3$ interfaces are formed by tuning the surface morphology of the BHJ film through a high-temperature annealing method. Compared to the as-cast OPD (control device), the typical dual-mode OPD can effectively detect the broad vis-NIR response range of 340–1000 nm with enhanced specific detectivity (D*) of -$10^{12}$ Jones in both PM and PV modes while exhibiting other satisfying performances, such as low operating bias (±2 V) and fast response speed (0.05/3.03 ms for PM-mode; 8.64/11.14 μs for PV-mode), simultaneously. In practical application, according to the incident light intensity, a suitable mode can be selected to keep the photocurrent stable at a relatively balanced level, avoiding signal amplification or high power consumption and thus reducing the burden of signal processing. Our work offers a generally applicable strategy to develop high-performance bias-switchable dual-mode OPDs, showing great potential in simultaneously meeting the application requirements of detecting dim and strong light and other variable application scenarios.

## Results

### Typical performance and working mechanism of dual-mode bias-switchable OPDs

The OPDs were fabricated with the classic inverted structure of ITO/ZnO/BHJ(PBDB-T:Y6)/$MoO_3$/Ag (Fig. 1a middle). The active layer was spin-coated at 100 °C to prevent PBDB-T aggregation as reported in our previous work[37]. Herein, to achieve PM effect, we annealed the 150-nm-thick BHJ active layers at a high temperature of 250 °C to alter the surface morphology and thus introduce structural traps at ZnO/BHJ

and BHJ/$MoO_3$ interfaces. The as-cast OPD (without active layer annealing) was prepared as a control.

In our design, inducing and blocking tunneling carriers under different bias directions by thermal-induced interfacial structural traps and blocking layers, are the key points to achieve dual-mode operation of the OPDs. The structural traps at ZnO/BHJ and BHJ/$MoO_3$ interfaces are introduced by altering the surface morphology of BHJ active layer by a high-temperature annealing method. The as-cast BHJ film exhibits a smooth and flat surface (Supplementary Fig. 1a) and has close contact with ZnO and $MoO_3$ layers, thus the carriers can easily pass through the ZnO/BHJ and BHJ/$MoO_3$ interfaces without hindrance (Fig. 1a top). On the contrary, the annealed BHJ film shall be rather rough, verified by the rough 3D morphology image measured by laser scanning confocal microscope (LSCM, Supplementary Fig. 1b). Numerous spatial blind alleys (referred as structural traps) will form at ZnO/BHJ and BHJ/$MoO_3$ interfaces, blocking the injection of electrons/holes from external circuit under forward bias and thereby reducing the $J_d$ (Fig. 1a bottom)[38]. Furthermore, as structural traps at interfaces have the same blocking effect on electrons and holes and can tune the trapped carrier type by bias direction, from the function perspective, they will also act as hole/electron traps when illuminating the dual-mode device under reverse/forward bias[39,40].

Figure 1b illustrates the working mechanism of the dual-mode device under different biases. (1) Under reverse bias, the device has low $J_d$ due to the large energy barriers from ZnO and $MoO_3$ blocking layers. When illuminated, the photogenerated electrons/holes in BHJ film are extracted by the ITO/Ag electrode to generate photocurrent, i.e. operating in PV-mode. In addition, since the transport and extraction of photogenerated carriers at interfaces are limited by the structural traps, the EQE values of the device in PV-mode are reduced as compared to that of the as-cast OPD (Fig. 1d). (2) Under forward bias, the energy barriers for charge injection from the electrodes are shallower, thus the $J_d$ is higher than that under reverse bias. When illuminated, almost all the photogenerated holes and electrons are trapped at the ZnO/BHJ and BHJ/$MoO_3$ interfaces respectively due to the disordered structural traps and the blocking of ZnO and $MoO_3$ layers. As the trapped and accumulational charges increase, the enhanced band bending finally leads to charge tunneling injection, and hence the dual-mode OPD operates in PM-mode under forward bias.

Figure 1c shows the current density–voltage (J–V) curves of as-cast and dual-mode OPDs. Due to the high dark current density ($J_d$) of the as-cast OPD under forward bias, the photoresponse cannot be effectively distinguished and thus it can only operate in PV-mode under reverse bias. After thermal annealing (30 min), the $J_d$ is significantly suppressed, especially under forward bias (-3 orders of magnitude lower), which makes it possible for the device to work under dual-mode operation. In the broad wavelength range of 340–1000 nm, the annealed device exhibits PM effect with EQE of $10^3$–$10^5$% under forward bias and maintains PV response under reverse bias (Fig. 1d). Owing to the suppressed $J_d$, the D* (shot-noise-limited, calculated by Supplementary Eq. 2) of the annealed device is improved in both PM and PV modes as compared to the control sample (Fig. 1e), indicating enhanced optoelectronic performance and multi-scenario application potential.

The trapped photogenerated carriers will also cause tunneling charge injection under reverse bias without the blocking layer. As shown in Supplementary Fig. 2, the $MoO_3$-free OPDs exhibit PM effect under both forward and reverse biases. Thus, the blocking layers play a vital role in maintaining PV response characteristic under reverse bias and ensuring the bias-switchability of operation mode. Furthermore, taking $MoO_3$ for example, the functions of blocking layer can be comprehensively summarized: (1) block the charge injection from external circuit in dark to reduce the dark current; (2) block the charge tunneling from external circuit under illumination to maintain PV-mode under reverse bias; (3) further enhance the accumulation of photogenerated holes at BHJ/$MoO_3$ interface under forward bias due

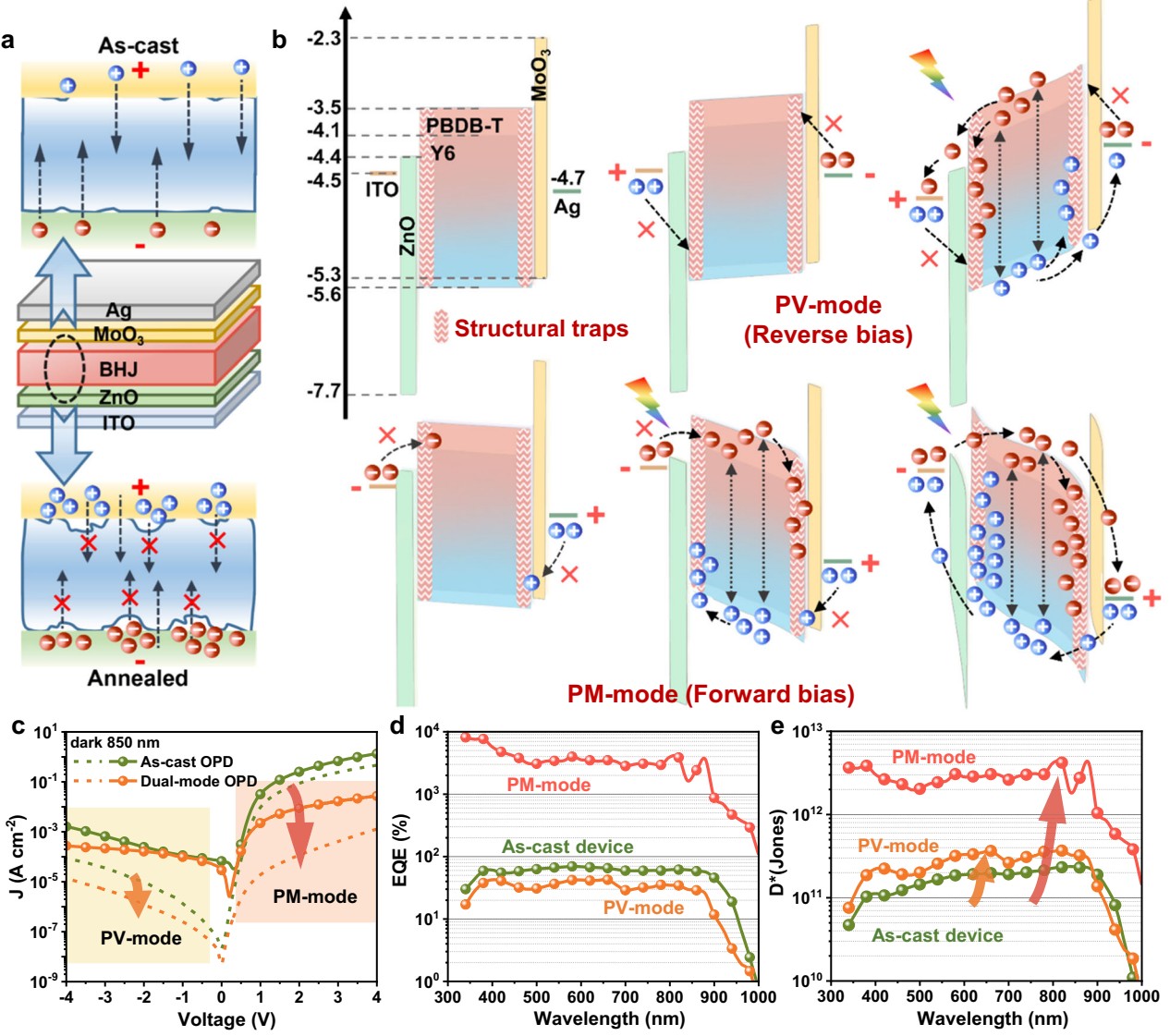

**Fig. 1 | Working mechanism and typical performance of dual-mode bias-switchable OPD. a** Schematic illustrations of the device structure (middle), as-cast (top) and annealed (bottom) ZnO/BHJ/MoO₃ interfaces under forward bias in dark. **b** Schematic diagrams illustrating the working mechanism of the dual-mode OPD operated in PV (reverse bias) and PM (forward bias) modes. **c** $J–V$ curves of as-cast and dual-mode devices in dark and under illumination (850 nm, 0.737 mW cm⁻²). **d** EQE spectra and **e** $D^*$ spectra of as-cast and dual-mode devices operated in PV (−2 V) and PM (+2 V) modes.

to its shallow lowest unoccupied molecular orbital (LUMO), and thus improve the PM effect.

## Evolution of BHJ films with different annealing durations and the effects on the interfacial band bending

PBDB-T:Y6 BHJ films were annealed at 250 °C for different durations and periodically characterized by optical microscopy (Fig. 2a) and LSCM (Supplementary Fig. 3). The as-cast BHJ film shows a flat and smooth surface. Micron-sized morphology coarsening appears when the film suffers from annealing, and the continuous process causes an increase in both quantity and size of the appeared aggregates. After 120 min, the aggregates can even grow to a particle size larger than the thickness of active layer, and embed throughout the entire active layer, which can also be observed from the bottom side (through the ITO glass, Supplementary Fig. 4). The aggregates growth process induced by continuous high-temperature annealing is consistent with the classical heterogeneous transformation process, *i.e.*, nucleation and growth[41–43]. Fewer and smaller aggregates are observed on the surface of thinner film (150-nm thick, the second

line of Fig. 2a) and no clear difference in morphology is observed from the other side. The surface morphology of the annealed BHJ films was also characterized by a 30°-tilted scanning electron microscope (SEM), and the evolution process is consistent with the content described above (Supplementary Fig. 5). To explore the composition of aggregates, PBDB-T:Y6 films with different proportions were prepared. After annealing at 250 °C for 30 min, the amount of aggregates present on the film surface increases significantly with the increase of Y6 content (Supplementary Fig. 6), proving that the aggregates were caused by Y6.

Recently, the cross-sectional scanning Kelvin probe microscopy (SKPM) has been employed to characterize the surface potential (SP) difference between functional layers in organic solar cells and photodetectors[28,44–46]. The vacuum level alignment within a device can be interpreted by multiplying the SP with electron charge −e, thus revealing the dynamic changes of interfacial band bending[46]. In order to further investigate the effect of BHJ morphology changes on interfaces, the *operando* cross-sectional SKPM measurements were carried out on the Ar⁺ beam milled edge under various operating conditions

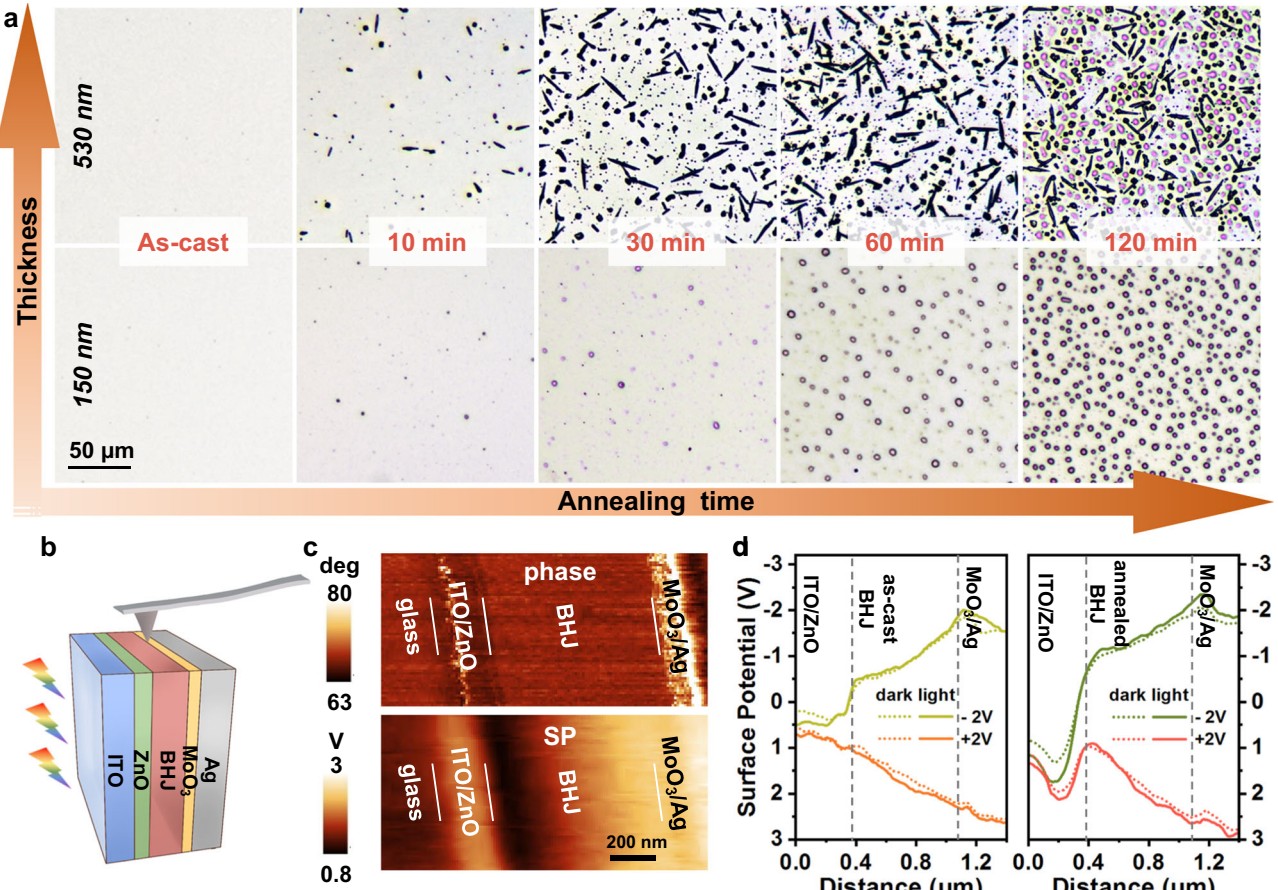

**Fig. 2 | Evolution of BHJ films with different annealing durations and the effects on the interfacial band bending. a** Optical micrographs for the top side of PBDB-T:Y6 BHJ films annealed at 250 °C for different durations. The thicknesses of BHJ films are 150 nm and 530 nm, as indicated in the left column. The scale bar (50 μm) is equal for all micrographs. **b** Schematic illustration of operando cross-sectional scanning Kelvin probe microscopy (SKPM) measurements during operating under conditions like illumination and bias voltages. **c** Phase and surface potential (SP) images of the 30-min-annealed device cross-section under illumination in PM-mode (+2 V) obtained by atomic force microscopy (AFM), the scale bar is 200 nm. **d** Surface potential depth profiles of the as-cast and 30-min-annealed OPDs in dark and under illumination (AM 1.5 G) at ±2 V.

(Fig. 2b). The thicknesses of BHJ films were increased to ~760 nm for clearer results.

The cross-sectional atomic force microscopy (AFM) phase images (Fig. 2c and Supplementary Fig. 7) show clear boundaries between the organic/inorganic layers in both as-cast and 30-min-annealed devices, which help identify the interfaces. Figure 2d shows the extracted SP profiles of as-cast and annealed devices in dark and under AM 1.5 G illumination at ±2 V. When the as-cast and annealed devices operate at the same bias, the overall SP profiles vary in the same direction, and the degree of SP variation increases under illumination, which is consistent with the working mechanism in Fig. 1b. But at the ZnO/BHJ and BHJ/MoO$_3$ interfaces, the band bending behavior of the two devices are quite different, especially in PM-mode (+2 V). Compared with as-cast device, when operating in PM-mode, rapid SP drop/rise at (ZnO/BHJ)/(BHJ/MoO$_3$) interfaces in annealed device are observed, indicating sharp upward/downward band bending, and vice versa in PV-mode. This can be attributed to the accumulation of photogenerated hole/electron at the interfaces[28], which demonstrates the existence of structural traps at both ZnO/BHJ and BHJ/MoO$_3$ interfaces in the annealed device.

**Performance of dual-mode bias-switchable OPDs with different annealing durations and BHJ thicknesses**

Optical absorption measurements of the 150-nm-thick BHJ films with different annealing durations were carried out and shown in Fig. 3a.

The spectra of the BHJ films displayed negligible changes under short-time annealing within 30 min. In contrast, continued annealing results in a substantial drop in the Y6 absorption region (i.e. above 700 nm) with almost no absorption by 120 min, which means the annealing time needs to be restricted. This can be interpreted as the gradual thermal decomposition of Y6 during the long-lasting high-temperature annealing process, since the absorption spectra of pure PBDB-T films are maintained (Supplementary Fig. 8a) and the absorption peak (850 nm) of pure Y6 films gradually decreases with the annealing time (Supplementary Fig. 8b). The effect of high-temperature annealing on the electrical property of the devices was investigated via electrochemical impedance spectroscopy (EIS) in dark condition (+2 V), and the Nyquist plots and equivalent electronic circuit are shown in Supplementary Fig. 9. The charge transfer resistance ($R_{ct}$), corresponding to the charge transfer between electrodes and BHJ layer at a low frequency, is markedly enhanced after annealing. The decrease in $R_{ct}$ with prolonged annealing process (over 30 min) can be attributed to the continuous growth of aggregates. The large-sized aggregates embed throughout the entire active layer, forming short circuit paths[41].

Due to the interfacial structural traps, the $J_d$ of annealed OPDs are effectively suppressed (Supplementary Fig. 10a). The decreasing open circuit voltage after annealing suggests the introduction of structural traps[47]. When operating in PM-mode (+2 V, Fig. 3b, c), the 60-min-annealed OPD can reach a high peak EQE of 43702% (@ 340 nm), whereas the device has similar $D^*$ values with 30-min-annealed one

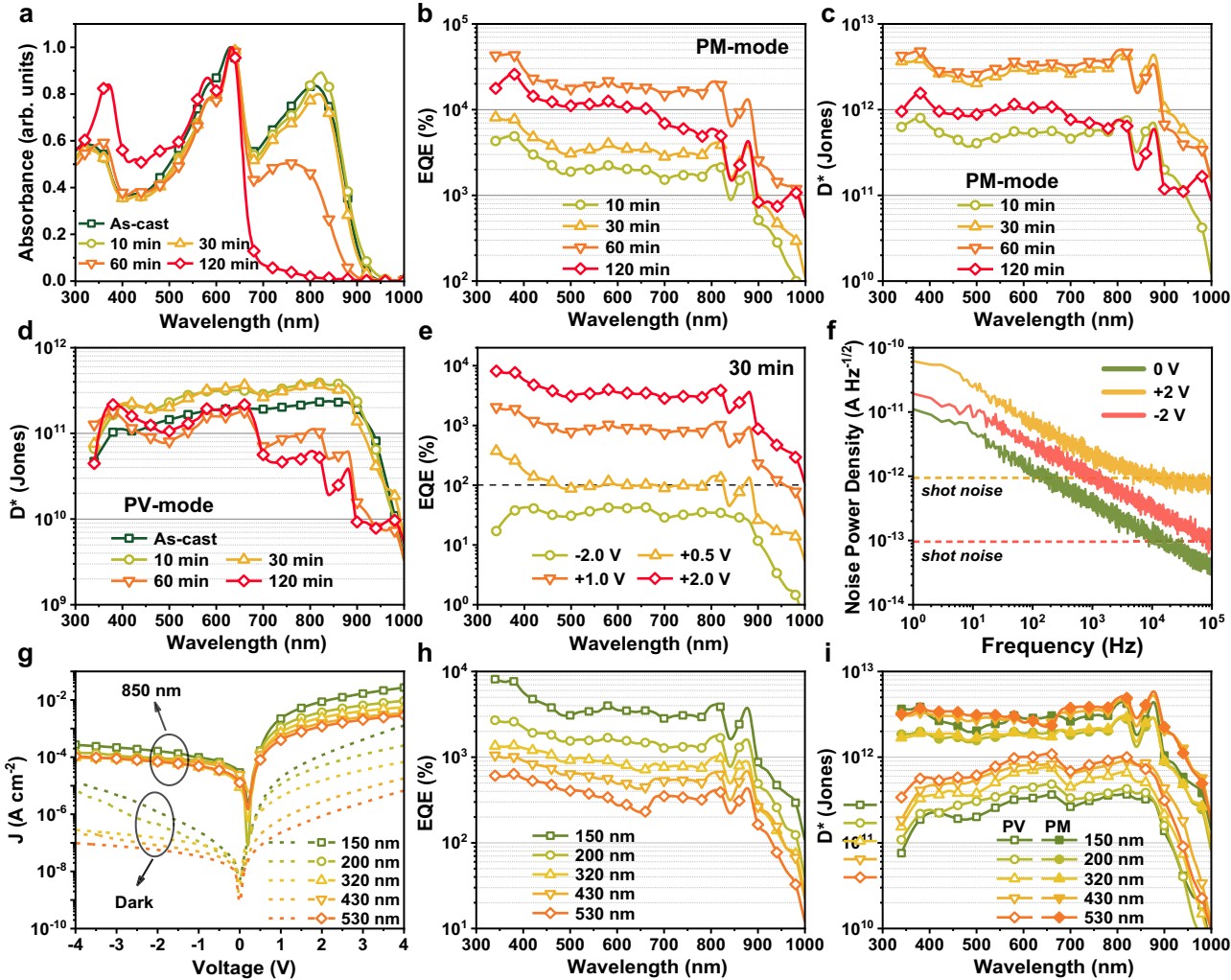

**Fig. 3 | Performance of dual-mode bias-switchable OPDs with different annealing durations and BHJ thicknesses. a** Normalized UV-vis-NIR absorption spectra of the BHJ films with different annealing durations. Performance of 150-nm-thick OPDs with different annealing durations: **b** EQE spectra and **c** $D^*$ spectra in PM-mode (+2 V). **d** $D^*$ spectra in PV-mode (−2 V). **e** EQE spectra of 30-min-annealed OPD under different biases. **f** Noise spectral density of 30-min-annealed OPD. Performance of 30-min-annealed OPDs with different BHJ thicknesses: **g** $J$-$V$ curves in dark and under 850 nm illumination (0.737 mW cm$^{-2}$). **h** EQE spectra in PM-mode. **i** $D^*$ spectra in PM and PV modes.

(EQE = 8133% @ 340 nm) due to the effect of $J_d$. Benefiting from reduced $J_d$, when operating in PV-mode (−2 V), OPDs annealed within 30 min show higher $D^*$ than that of the as-cast and 60-min-annealed devices (Fig. 3d). The peak EQE values are 71% and 43% for as-cast and 30-min-annealed devices at −2 V, respectively (Supplementary Fig. 10b). The reduced EQE in PV-mode after annealing can be attributed to: (1) the high-temperature annealing process introduces a large number of structural traps at ZnO/BHJ and BHJ/MoO$_3$ interfaces, limiting carrier transport and collection; (2) bimolecular recombination is intensified in BHJ by poor charge transport. However, longer-time-annealed samples exhibit reduced on-off ratios (under 850 nm illumination) and much lower $D^*$ in the NIR region, consistent with the attenuation of spectral absorption. Considering the photoresponse to the entire vis-NIR region in both PM and PV modes, the 30-min-annealed OPD exhibits the best optoelectronic performance. Figure 3e demonstrates that the 30-min-annealed OPD only requires a low forward bias of +0.5 V to produce PM effect, and EQE values gradually increase with the applied forward bias.

The noise spectral density ($S_n$) of the optimized OPD in the test frequency range of 1 Hz−100 kHz (Fig. 3f) shows that $S_n$ is dominated by $1/f$ noise in low-frequency range and shot noise in high-frequency range. The device has relatively rising $S_n$ in PM-mode, which is

consistent with the $J_d$−$V$ characteristic. Whereas, a lower noise equivalent power (NEP) of $10^{-13}$ W/Hz$^{1/2}$ was measured in PM-mode (Supplementary Fig. 11a) at the modulation frequency of 60 Hz owing to the higher EQE. The maximum $D^*$ values derived from $S_n$ (Supplementary Fig. 11b, calculated by Supplementary Eq. 1) are $1.1 \times 10^{10}$ Jones in PV-mode (820 nm) and $4.5 \times 10^{11}$ Jones in PM-mode (880 nm).

The durability of as-cast and dual-mode OPDs (150 nm-thick, 30 min-annealed, $N = 5$) was investigated by placing them in air environment at room temperature for two months (Supplementary Fig. 12). The EQE and $D^*$ values of all the devices have different degrees of attenuation. In contrast, the performance attenuation of dual-mode OPDs in both PM and PV modes is slightly higher than that of as-cast ones. This can be attributed to the increased degradation of organic active layer materials after high-temperature annealing, especially for Y6, thus adversely affecting the durability of the dual-mode device.

To investigate the effect of BHJ thickness on device performance, a series of 30-min-annealed OPDs with BHJ thickness ranging from 150 to 530 nm were fabricated. The $J_d$ and EQE of the OPDs in both two operating modes decrease significantly and regularly with the BHJ thickness (Fig. 3g, h and Supplementary Fig. 13 for EQE spectra in PV-mode). Since EQE in PV-mode does not change in the order of

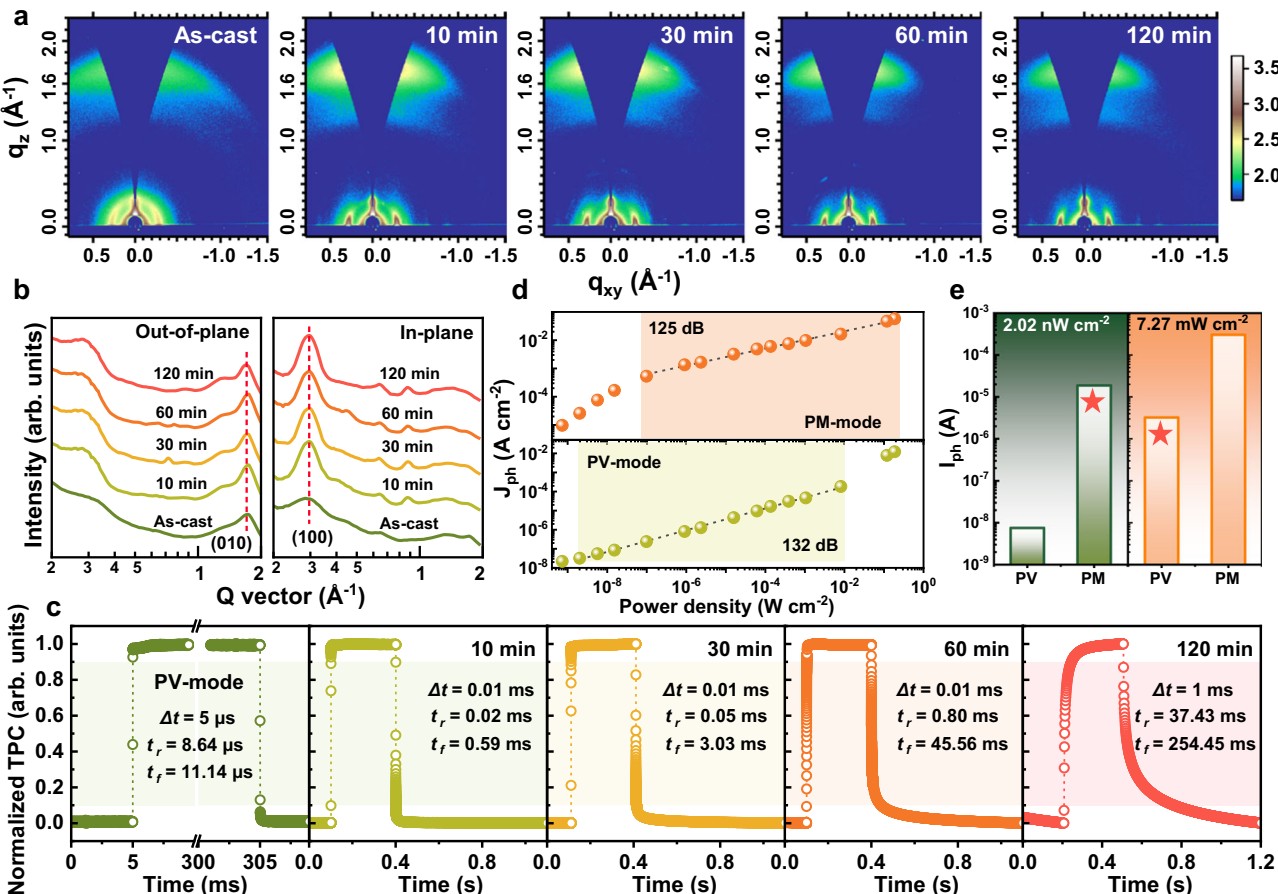

**Fig. 4 | Grazing-incidence wide-angle X-ray scattering (GIWAXS), power dependence and response time analysis of dual-mode bias-switchable OPDs.** **a** 2D GIWAXS images of the BHJ films with different annealing durations. **b** The corresponding line-cut profiles of out-of-plane and in-plane directions. **c** Transient photocurrent (TPC) curves of 30-min-annealed device in PV-mode (−2 V), and devices with different annealing durations in PM-mode (+2 V) under incident 850 nm optical signal. **d** Power dependence of the 30-min-annealed device in PM and PV modes under 850 nm illumination. **e** Photocurrent of the 30-min-annealed device in different modes under different incident 850 nm light intensities.

magnitude as in PM-mode, the reduced $J_d$ has a greater effect on $D^*$ and thus obviously improved $D^*$ values are obtained in PV-mode after increasing BHJ thickness (Fig. 3i). The full set of performance of the 530-nm-thick OPDs with different annealing durations has also been systematically studied, including $J$-$V$ curves, EQE and $D^*$ spectra in PM and PV modes (Supplementary Fig. 14). Overall, the 30-min-annealed 530-nm-thick OPD has comprehensively good performance of exceeding $10^{12}$ Jones in both two modes, respectively. Further, the 530-nm-thick OPD has lower $S_n$, but due to the suppressed EQE, the NEP, and corresponding $S_n$-derived $D^*$ values are similar to those of the 150-nm-thick device (Supplementary Fig. 15).

## GIWAXS, power dependence and response time analysis of dual-mode bias-switchable OPDs

The 2D Grazing-incidence wide-angle X-ray scattering (GIWAXS) images and corresponding line-cut profiles of BHJ films annealed for different durations are shown in Fig. 4a, b. Clear (010) $\pi$-$\pi$ stacking diffraction peaks ($q_z \approx 1.71$ Å$^{-1}$) in the out-of-plane (OOP) direction and (100) lamellar stacking peaks ($q_{xy} \approx 0.29$ Å$^{-1}$) in the in-plane (IP) direction can be observed in those 2D images, especially for the annealed films, revealing highly oriented face-on structure[48,49]. For more quantitative analyses, the stacking distance ($d$), crystalline coherence length ($L_C$, calculated by Scherrer equation), and paracrystalline disorder parameters $g_{(010)}$ and $g_{(h00)}$ are summarized in Supplementary Table 1. After annealing, the $L_C$ of BHJ films increased significantly and the $\pi$-$\pi$ stacking $L_C$ reached the maximum at 30 min, which indicates higher

regularity and crystallinity of annealed films and is beneficial to improving the carrier mobility.

The paracrystalline disorder parameter for $\pi$-$\pi$ stacking in the OOP direction can be calculated by

$$g_{(010)} = \sqrt{\frac{\Delta q}{2\pi q_0}} \tag{2}$$

where $\Delta q$ and $q_0$ are the width and center position of the diffraction peak, respectively[50]. The paracrystalline disorder parameter for lamellar stacking in the IP direction can be calculated from the slope ($m$) of $\delta b$–$h^2$ plot (Supplementary Fig. 16), which is determined by

$$m = \frac{g_{(h00)}^2 \cdot \pi^2}{d} \tag{3}$$

where $\delta b$ is the integral width of the diffraction peak, $h$ is the order of diffraction and $d$ is the domain spacing[51]. The $g_{(010)}/g_{(h00)}$ of the BHJ films annealed for 0–120 min are 14.47/21.75%, 12.03/10.26%, 11.87/9.68%, 12.04/9.73%, and 12.39/10.61%, respectively. Collectively, lower paracrystalline disorder parameter values are obtained for the annealed films compared to those of as-cast film, and the value is the smallest for the 30-min sample. This trend implies apparently lower paracrystalline disorder and structural defect states in annealed BHJ film, leading to high charge carrier mobility and suppressed electron detrapping from defects when operating in PM-mode[27,34]. In addition,

the time-resolved photoluminescence (TRPL) spectra (Supplementary Fig. 17) show reduced average lifetime $\tau_{avg}$ of 30-min-annealed BHJ film, which indicates decreased defect density in the BHJ bulk and thus suppressed the relatively slower trap-assisted recombination, consistent with the GIWAXS results[52].

Benefiting from this, the devices exhibit high response speed in both PV and PM modes, the transient photocurrent (TPC) curves of the devices under incident 850 nm optical signal with a frequency of 1 Hz and duty ratio of 30% are shown in Fig. 4c. When operating in PV-mode, the devices usually achieve a fast response time (defined as the 10–90% duration time of the maximal photocurrent) in the microsecond range because there is no time-consuming charge accumulation process. For the typical 30-min-annealed device, the fast rise time ($t_r$) and fall time ($t_f$) of 8.64 μs and 11.14 μs are obtained in PV-mode at −2 V, which are slightly slower than the as-cast OPD (4.15/4.19 μs, −2 V, Supplementary Fig. 18) due to the introduced structural traps after annealing. Generally, due to the slow charge-accumulation and band-bending process, the response speed of reported PM OPDs is much slower and generally in the order of milliseconds or even seconds. In this work, the $t_r/t_f$ of the annealed devices in PM-mode (+2 V) gradually increase with the annealing time and the typical 30-min-annealed device showed 0.05/3.03 ms, which is among the fastest speeds in the recently reported PM OPDs (Supplementary Table 2). Besides the optimized face-on packing orientation and lower paracrystalline disorder mentioned above, the device also maintains the advantages of high-speed dissociation and transport of photogenerated excitons in BHJ film, accelerating the charge accumulation and thus shortening the $t_r/t_f$ in PM-mode[53]. The growing trend of $t_r/t_f$ along with annealing time is mainly attributed to the positive correlation between response time and EQE/gain, as higher EQE/gain means more charge filling and releasing processes. The large difference between $t_r$ and $t_f$ in PM-mode can be explained as the quick trap-filling process when light incident and the tardy recombination of the trapped electron by injected holes when light is off. In addition, the RC time constant of the device is also an important factor limiting the response speed, which is determined by the product of series resistance ($R_s$) summation and capacitance[10]. Along with the annealing time, the rougher surface morphology of BHJ films leads to a larger contact resistance of the device and thereby increases the $R_s$. Therefore, the TPC curve of 120-min-annealed device displays a distinct large RC time constant waveform with significantly increased $t_r$ and $t_f$. Additionally, similar regularities have also been verified by 530-nm-thick devices (Supplementary Fig. 19), and the devices exhibit relatively fast response speed since the thicker BHJ film leads to low capacitance and therefore rather short RC time[16].

The transient photoresponse behavior of the 150-nm-thick 30-min-annealed OPD under continuous pulse signal (850 nm) with a duty ratio of 30% was measured (Supplementary Fig. 20). When operating in PV-mode, the output of device can follow the on-off switching of the incident signal and achieve steady state dark current and photocurrent even at the frequency of 1 kHz. When operating in PM-mode, the response begins to decrease at 1 kHz and cannot fully reach/decay to the original photocurrent/dark current due to the reduced response speed. Furthermore, the −3 dB cutoff frequency, defined as the frequency at which the output of device is attenuated to 0.707 of the original amplitude, of the OPDs were measured under 850 nm light (Supplementary Fig. 21). Both the as-cast OPD and dual-mode OPD (PV-mode) exhibit similar -3dB cutoff frequencies of exceeding 100 kHz (153.1/113.5 kHz) due to the fast response speed, and the as-cast one has slightly larger frequency due to the faster response speed. Owing to the time-consuming carrier accumulation and band tunneling processes in PM effect, the -3dB cutoff frequencies of dual-mode OPD operating in PM-mode and 30-min-annealed MoO₃-free OPD (+2 V) both decrease significantly to 8.9 kHz and 20.8 kHz respectively. The shallow LUMO level of MoO₃ allows dual-mode OPD to accumulate more carriers at the interface and thus obtain higher EQE, but this also

requires a longer accumulation time and therefore has a lower -3dB cutoff frequency than MoO₃-free OPD.

To further characterize the 30-min-annealed device, the incident light power ($P_{in}$) dependence of the photocurrent density ($J_{ph}$) was measured in both PV and PM modes using an NIR light-emitting diode (LED, 850 nm) over the $P_{in}$ ranging from ~$10^{-9}$ to $10^{-1}$ W cm$^{-2}$ (Fig. 4d). In this double logarithmic diagram of $J_{ph}$–$P_{in}$, $J_{ph}$ presents a sublinear relationship with the incident light intensity in the range of 132 dB (PV-mode) and 125 dB (PM-mode), and the 530-nm-thick device shows similar characteristic (Supplementary Fig. 22). Different from the linear dynamic range (LDR, slope = 1), here the $J_{ph}$–$P_{in}$ curves exhibit sublinearity with slope <1. This is due to the introduction of structural traps at the interfaces, which leads to the hindering of photogenerated carrier transport. Thus the bimolecular recombination will gain favorable competition against the extraction and collection of photogenerated carriers in BHJ, resulting in a sublinear $J_{ph}$–$P_{in}$ dependence[54,55]. In contrast, the as-cast OPD without structural traps shows a linear $J_{ph}$–$P_{in}$ dependence of 141 dB, while the 30-min-annealed MoO₃-free OPD is similar to that of dual-mode OPD in PM-mode, displaying a sublinear relationship of 119 dB (Supplementary Fig. 23), which is a typical feature of PM-type OPDs[26,51,56,57]. The LDR is the preferred performance parameter, but the sublinear characteristic in the double logarithmic diagram can be processed by mature data fitting[57]. In addition, the high $R$ at low light intensity also has potential applications such as low light detection or imaging.

Figure 4e displays the photocurrent in different modes under different incident light intensities. It can be intuitively seen that PM-mode at low light intensity (2.02 nW cm$^{-2}$) and PV-mode at high light intensity (7.07 mW cm$^{-2}$) can keep the photocurrent stable at a relatively balanced level. Therefore, a suitable mode can be selected according to the incident light intensity in practical application. For dim scenarios like night vision, the PM-mode with high EQE can avoid the use of amplifying circuits; for the strong-light scenarios like under direct sunlight, the low-power-consumption PV-mode can avoid potential damages (heat-dissipation, breakdown, etc.) to the organic active layer. This is beneficial for reducing costs and the burden of signal processing.

## Generality of the high-temperature annealing method for fabricating dual-mode bias-switchable OPDs

To validate the general applicability of the high-temperature annealing method for fabricating the dual-mode bias-switchable OPDs, the method was applied to other organic BHJ films, i.e. PM6:Y6 and PBDB-T:ITIC-Th, and the ~300-nm-thick BHJ film was annealed at 250 °C for 60 min and 30 min, respectively. The energy level diagrams of the OPDs are shown in Fig. 5a, d. Compared with the as-cast device, the PM6:Y6 and PBDB-T:ITIC-Th OPDs both have reduced $J_d$ (Fig. 5b, e), and exhibit PM/PV effect at forward/reverse bias, respectively (Fig. 5c, f), verifying the bias-switchable dual-mode performance. Supplementary Fig. 24 indicates the two devices exhibit $D^*$ values of reaching $10^{12}$ Jones in both modes. Therefore, the mechanism and realization method can be generally applied to OPDs with other organic BHJ active layers.

## Discussion

In conclusion, we have demonstrated a series of dual-mode OPDs exhibiting bias-switchable PM and PV operating characteristic. The structural traps introduced at ZnO/BHJ and BHJ/MoO₃ interfaces by high-temperature annealing method cause charge tunneling injection to obtain PM effect under forward bias, while the blocking layer prevents tunneling electron injection to maintain PV effect under reverse bias. The working mechanism of dual-mode OPD is confirmed by *operando* cross-sectional SKPM measurements. In the broadband range of 340–1000 nm, the typical 30-min-annealed OPD (active layer thickness of 150 nm) performs higher $D^*$ than the as-cast device in both PM and PV modes. Fast response speed is obtained in both PM (0.05/

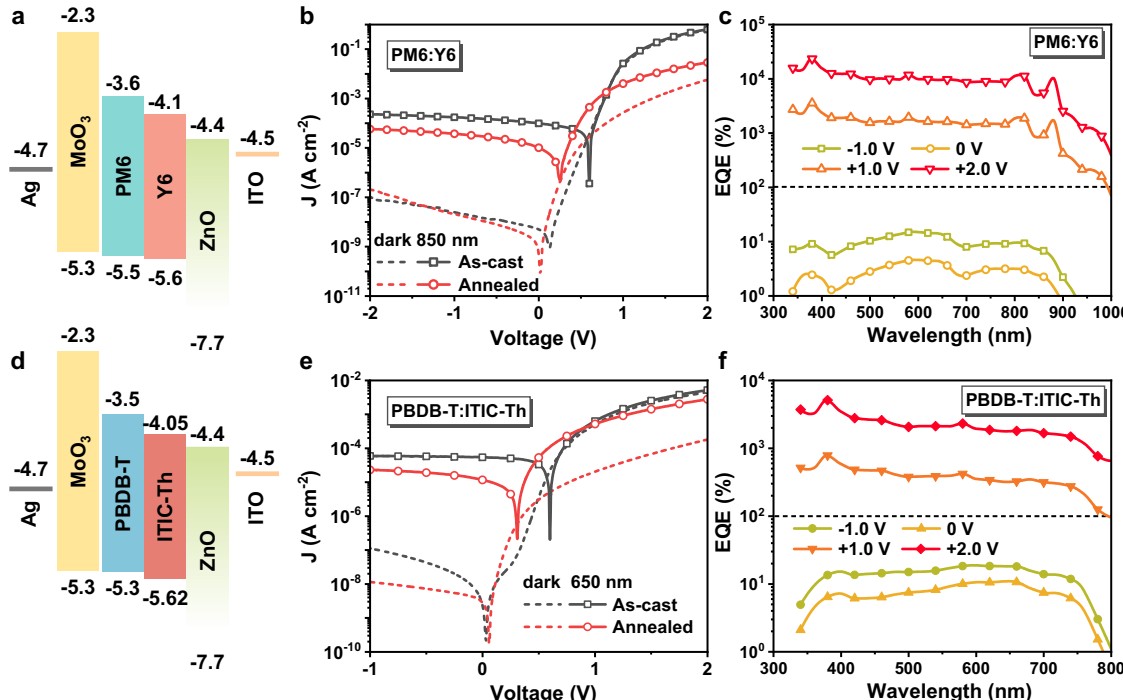

**Fig. 5 | Generality of the high-temperature annealing method for fabricating dual-mode bias-switchable OPDs.** Performance of the dual-mode OPDs based on PM6:Y6 and PBDB-T:ITIC-Th BHJ active layers, and the BHJ films were annealed at 250 °C for 60 and 30 min, respectively. **a, d** Energy level diagrams of the OPDs. **b, e** J–V curves in dark and under illumination. **c, f** EQE spectra of the devices under different biases. It can be observed that the operating mode of the PM6:Y6 and PBDB-T:ITIC-Th OPDs can also be switched by the bias direction.

3.03 ms) and PV (8.64/11.14 μs) modes owing to the lower para-crystalline disorder and defect states in BHJ film. Moreover, the effects of annealing duration and BHJ thickness on device performance are systematically discussed, and the general applicability of the realization strategy has been validated on other OPDs. Furthermore, the OPDs are promising in simultaneously meeting the application requirements of dim and strong light, avoiding signal amplification or high power consumption, and thus reducing the burden of signal processing.

We presume that high-performance bias-switchable dual-mode devices should meet the following requirements: based on the typical BHJ diode-type OPD, (1) the charge injected from the metal anode should be blocked to reduce the dark current under forward bias. The possible strategies include tuning the BHJ/electrode interface contact characteristics, or/and using a blocking layer. (2) Structural traps should be introduced in the BHJ or at the BHJ/electrode interface to produce PM effect under illumination. Besides the high-temperature annealing method adopted in this paper, other controllable/designable approaches, such as nano-imprinting, laser processing, template growth, etc., can be used to tune the surface micro-nano structure of BHJ film and thus introduce structural traps. In addition, a similar effect can be achieved theoretically by trapping/blocking photogenerated carriers using a designed interfacial layer[53]. (3) Blocking layers on both sides of the BHJ are essential to maintain the PV characteristics under reverse bias, although it may sacrifice part of the EQE in PM-mode. Our work opens a door for new type multi-mode OPDs with a generally applicable realization strategy and validated working mechanism. This device can also be developed to meet the demands of variable application scenarios.

## Methods

### Materials
The organic BHJ materials (PBDB-T, Y6, PM6, and ITIC-Th) were purchased from Solarmer Materials Inc. Zinc acetate dihydrate, ethanolamine, 2-methoxyethanol, chlorobenzene and 1-chloronaphthalene were purchased from Sigma-Aldrich. MoO₃ and Ag were from ZhongNuo Advanced Material (Beijing) Technology Co. Ltd. ITO substrates with a sheet resistance of <15 Ω sq⁻¹ were purchased from South China Xiangcheng Technology Co., Ltd.

### Device fabrication
The OPDs were fabricated with an inverted architecture of ITO/ZnO/PBDB-T:Y6/MoO₃/Ag. The transparent glass substrates with patterned ITO were scrubbed with detergent and then sonicated in deionized water, acetone and isopropanol subsequently and then dried by nitrogen stream. The cleaned substrates were treated with UV-Ozone for 20 min before use. The ZnO precursor solution (dissolving zinc acetate dihydrate in a mixture of ethanolamine and 2-methoxyethanol) was spin-coated on the ITO at 4000 rpm for 40 s, and then annealed at 200 °C for 30 min under atmosphere to obtain the ~30 nm ZnO layer. The PBDB-T:Y6 (1:1.2) blends were fully dissolved in chlorobenzene:1-chloronaphthalene (CB:1-CN, 199:1, v/v) at total weight concentrations of 30 mg mL⁻¹ (for typical 150 nm BHJ film). Before spin-coating, the blend solution and substrates were preheated on a hot plate at 100 °C. The active layer thickness was controlled by changing the solution concentration and spin-coating speed. The active layers were then thermally annealed at 250 °C for different durations under nitrogen atmosphere. The as-cast OPD (without active layer annealing) was prepared as control. Finally, the MoO₃/Ag (10/100 nm) was sequentially thermally evaporated onto the active layer under high vacuum (<10⁻⁴ Pa).

### Measurement and characterizations
The active layer thickness was determined by Ambios XP-300 surface profiler. The active layer morphologies were characterized by optical microscope (Leica Microsystems CMS GmbH), laser scanning confocal microscope (OLYMPUS OLS3100), and SEM (ZEISS GeminiSEM 300). The vis-NIR absorption spectra were measured by SHIMADZU UV-

3600. The EIS spectra were tested by an electrochemical workstation (CHI660D, CH Instruments). The current-voltage curves, $R$ and EQE spectra were measured with a semiconductor characterization system (Keithley 4200) under a Xenon light source coupled with a monochromator, calibrated with a standard Si detector (S1337-1010BQ, Hamamatsu Photonics). The smooth device cross-sections were prepared by cutting the devices from the back of glass substrate, and then milled the exposed edge by $Ar^+$ beam (Ilion$^+$II 697, Gatan Inc.) under vacuum for 2 h. The phase and SKPM SP images of device cross-section were obtained by a Cypher S AFM (Asylum Research, Oxford Instruments) in Ar-filled glovebox, and the light source was AM 1.5 G solar simulator light transmitted by full-spectrum optical fiber. The noise spectral density and response time were measured by a semiconductor parameter analyzer (FS-Pro, Primarius Technologies). GIWAXS measurements were conducted by Xeuss SAXS/WAXS system. TRPL measurements were carried out by a fluorescence lifetime spectrometer (FluoTime 300, PicoQuant). For power dependence measurement, a series of filters were used to modulate the incident light intensity while an LED of 850 nm was used as a light source.

## Data availability
The data that support the findings of this study are available in the Figshare database, at https://figshare.com/s/d8597885cafb6ef434b8. All other data are available from the corresponding author upon request.

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

## Acknowledgements

This work is supported by the National Science Fund for Distinguished Young Scholars (Grant No. 62225106 received by H.L.), National Natural Science Foundation of China (Grant No. U19A2070 received by Y.J. and Grant No. 22105032 received by L.Y.), and Sichuan Province Science and Technology Support Program (No. 2021YFH0186 received by Y.W.). The authors are grateful to Dr. Jianqi Zhang (National Center for Nanoscience and Technology, Beijing) for his help in GIWAXS measurement. The authors are especially grateful for technical support on SKPM and AFM measurements from Dr. Qi Chen and Ms. Ni Yin (Suzhou Institute of Nano-Tech and Nano-Bionics, Chinese Academy of Sciences, Suzhou). We have also benefited from very fruitful discussions with Professor Yang Chai (Department of Applied Physics, The Hong Kong Polytechnic University, Hong Kong).

## Author contributions

Q.L. and Y.W. conceived the idea and designed the experiments; Q.L., L.L., and J.W. performed the major experiments and aggregated the figures; L.Y., Z.J., and J.X. performed the characteristic analysis; D.G., W.L., and H.T. assisted with the theory study; Q.L. wrote the draft; Y.W., Z.J., and Y.J. revised the manuscript; all authors discussed the experimental results and commented on the manuscript.

## Competing interests

The authors declare no competing interests.
