## [Peer Review File · Nature Communications]

Organic Photodiodes with Bias-Switchable Photomultiplication and Photovoltaic ModesREVIEWER COMMENTS

Reviewer #1 (Remarks to the Author):

The authors proposed bias-switchable organic photodiodes by introducing traps at BHJ/MoO₃ interface. They demonstrated the good performance of the annealed devices in both PV and PM modes by applying different bias voltages. The working mechanism of dual-mode devices is discussed systematically. The novel design and the interesting results are impressive. However, some issues should be addressed before considering the acceptance of the manuscript.

1. In the Abstract, the authors mention that PV-type photodiodes have limited sensitivity and PM photodiodes need high power consumption. I am wondering what exactly are the advantages of the dual-mode photodiodes, since their sensitivity is still limited in PV-mode and bias is still required in PM-mode.
2. Why are traps concentrated at the BHJ/MoO₃ interface?
3. The authors should clarify the composition of the aggregates and explain why the aggregates form traps rather than acting as insulating islands.
4. The description of Fig. 4 f-i is missing.
5. PV-type photodiodes feature fast response to light signals. But the markedly increased trap density in the annealed device is supposed to increase the response time. Furthermore, the EQE of the annealed device working in PV-mode is quite low. Therefore, it is recommended that the provided device characteristics be discussed in detail with the reported values to better understand the effect of annealing on device performance.
6. The authors should add more experimental results to demonstrate the “non-close-contact BHJ/MoO₃ interface”.
7. The results indicate long-time annealing caused the decomposition of Y6 which represents the destruction hole transport channels. Accordingly, over-heating may favor hole blocking in PM-mode devices and retard hole transport in PV-mode devices. This trade-off should be discussed. In addition, Also, because of this trade-off, the authors are advised to state whether there are ways to further improve the performance of the device.

Reviewer #2 (Remarks to the Author):

The purpose of this work is to address the issues of the limited sensitivity of photovoltaic (PV)-type organic photodiodes under dim light and the limited stability of photomultiplication (PM)-type organic photodiodes under strong light. The author proposed an organic photodiode (OPD) that has a PM detection mode when it is operated under a forward bias, and it also acts as a PV-type OPD when it is operated under a reverse bias. The bias-switchable PM- or PV-mode detection in one OPD was realized by introducing thermal-induced interfacial traps and charge blocking layers at the bulk heterojunction (BHJ)/MoO₃ interface. The combined effects lead to the unique PM- and PV-type OPD detection behaviors: e.g., (1) reduced dark current and thereby improved detectivity; (2) enhanced carrier mobility and thus faster response speed of OPD when operating in the PV-mode. (3) high EQE due to the tunneling charge injection in the OPD operated under PM mode (forward bias). The results are interesting. However, the PM-mode OPD shows better photoresponse linearity under illumination of a strong light than that of the PV-mode OPD, while its PV-mode has a better photocurrent linearity under illumination of a weak light than that of its PM-mode, as indicated in Fig 4c. The performance comparisons between single-mode PM-type OPD, single-mode PV-type OPD and the dual-mode OPD are not presented in the manuscript. The aim of the work as highlighted above is not supported fully by the results presented in the present format of the manuscript.

The following issues should be addressed before the revision can be considered for a possible publication in Nature Communications:

1. The light power dependence of the photocurrent density measured for the single-mode PM-type OPD and single-mode PV-type OPD should be provided for comparison studies, e.g., compare the results obtained for the dual-mode OPD that is operated under the corresponding PM- and PV- modes. Please discuss the advantages and disadvantages of the dual-mode OPDs compared to that of the single mode OPDs.
2. Please check the figure captions in Fig 4.
3. What is the mechanism accounting for the reduced dark current in the annealed OPDs operating in either PM or PV mode?
4. The appearance of the microscopic aggregation is usually associated with the degradation in OPDs, which is observed in the thermal annealed OPDs as presented in the manuscript. The author shall discuss the durability of the dual-mode OPDs as compared to that of the as-casted devices.

5. What is the meaning of “the aggregates that continuously grow beyond the film thickness” in line 185-186, on page 9?

6. A reverse bias of -40 V was applied to the OPD, as indicated in supplementary Fig 2c. Please explain why such a high voltage is required for the operation of the OPDs?

7. What are the -3dB cut-efficiencies measured for the dual-mode OPD operated under its PM- and PV-modes? Are the -3dB cut-efficiencies measured for the dual-mode OPD operated under its PM- or PV-mode similar to that obtained for the corresponding desecrate PM and PV OPDs?

8. Did the thermal-induced interfacial traps occur only at the BHJ/MoO₃ interface or the traps also formed at the ZnO/BHJ interface in the dual-mode OPD? If the traps exist at both BHJ/MoO₃ and ZnO/BHJ interfaces, which interface plays a dominant role in determining the PM behavior in the dual-mode OPD?

Reviewer #3 (Remarks to the Author):

This is the reviewer report for then manuscript entitled “Organic Photodiodes with Bias-Switchable Photomultiplication and Photovoltaic Modes” for Nature Communications.

In brief, the authors of this paper advance organic photodiodes based on Bulk Heterojunction (BHJ) devices which operate in a photovoltaic (PV) mode or a photomultiplication (PM) mode depending on the polarity of the applied voltage. Specifically, by applying a forward bias the devices show a PM effect resulting in very large EQEs, while the “normal” PV mode is obtained when device is operated in the reverse bias. This behaviour is obtained by conducting an annealing treatment of the BHJ at a relatively high temperature, and is attributed to the formation of thermal-induced traps at the BHJ/MoO₃ interface.

While the idea and concept are both very interesting and the experimental results looks promising, my main criticism is that the proposed working principle and physical mechanism behind the PM effect in forward bias seems lacking and not supported by convincing experimental evidence.

Detailed comments:

1. Traps are usually either acceptor-type or donor-type, meaning that they will either be electron traps or hole traps, respectively. An acceptor-type (electron) trap will be neutral if empty and negatively charged when occupied by an electron, whereas a donor-type trap is positively charged when empty (i.e., occupied by a hole) and neutral when occupied by an electron. In this work, however, it is suggested that the traps can act as both electron and hole traps, simply based on the applied bias. How can this be the case? Such claims require more theoretical and experimental justification.

2. Why are thermal-induced traps only formed at the BHJ/MoO₃ interface?

3. The authors use SCLC to estimate the trap density in both as-cast and annealed devices (Fig 3f). However, Eq. (2) assume that the traps are distributed throughout the bulk of the active layer, rather than interfacial traps.

4. It is not clear why the J_{ph} has a sublinear dependence with P_{in} in the relevant regime. Since trap-assisted recombination is usually considered a (nearly) first-order recombination process, I would have expected the J_{ph} vs P_{in} to display a (almost) linear dependence. This is not seen for either PV or PM mode. In fact, the J_{ph} vs P_{in} curves in the PV mode (in Fig 4c) seem to show close to a slope of 1/2 on the log-log scale, which is what one would be expected if the current is strongly limited by bimolecular recombination. This would suggest that there is something wrong/missing in the working mechanism proposed by the authors?

5. Based on GIWAXS, the authors conclude that there is a higher regularity and crystallinity of annealed films (beneficial for improving the mobility). These findings seem contradictory to the other experimental data that i) the EQE in PV mode is lower compared to as-cast device, and ii) the suggestion that there is a large number of traps and worse surface morphology in annealed devices. Also, the sublinear J_{ph} vs P_{in} curve is consistent with poor charge transport, suggesting a low mobility in the annealed devices.

6. Finally, I also got confused about the charge transfer resistance R_{ct} that the authors extracted from impedance spectroscopy. The increased R_{ct} is attributed to restricted injection in annealed devices. However, the R_{ct} in Fig 2d is approximately only increased by a factor of 3 compared to as-cast case. I fail to see how such a relatively small change can give rise to reduction of J_d in the forward-bias by several orders of magnitude?

Response to reviewers

TITLE: Organic Photodiodes with Bias-Switchable Photomultiplication and Photovoltaic Modes

(Nature Communications manuscript NCOMMS-22-53601A-Z)

Dear reviewers,

Thank you all very much for your time and valuable suggestions for improving the quality of our manuscript submitted to your esteemed peer reviewed journal. The list of point-by-point response to the reviewers' comments is provided below. All corrections/modifications made in the revised manuscript were highlighted with **Yellow Color**.

Reviewers' comments:

Reviewer #1:

The authors proposed bias-switchable organic photodiodes by introducing traps at BHJ/MoO₃ interface. They demonstrated the good performance of the annealed devices in both PV and PM modes by applying different bias voltages. The working mechanism of dual-mode devices is discussed systematically. The novel design and the interesting results are impressive. However, some issues should be addressed before considering the acceptance of the manuscript.

Response: We appreciate the reviewer's commendatory comments and suggestions to further improve the quality of this work. The response to specific comments is provided below.

1. In the Abstract, the authors mention that PV-type photodiodes have limited sensitivity and PM photodiodes need high power consumption. I am wondering what exactly are the advantages of the dual-mode photodiodes, since their sensitivity is still limited in PV-mode and bias is still required in PM-mode.

Response:

Thanks for the comments. The single-mode PV/PM-type devices can achieve good device performance, but still they have limitations in certain application scenarios. The advantage of the dual-mode device is that it can choose the corresponding operating mode for different application scenarios to keep the photocurrent stable at a relatively balanced level. For the dim scenarios like night vision, the PM-mode with high EQE can avoid the use of amplifying circuits; for the strong-light scenarios like under direct sunlight, the low-power-consumption PV-mode can avoid potential damages (heat-

dissipation, breakdown, etc.) to the organic active layer. This is beneficial for reducing costs and the burden of signal processing.

2. Why are traps concentrated at the BHJ/MoO₃ interface?

Response:

This is an inspiring comment.

In the previous manuscript, we have carried out a series of characterization on the top surface (adjacent to MoO₃) of the active layer (including optical microscopy, SEM and laser scanning confocal microscope), and obtained obvious morphological changes. However, since it is very difficult to peel off the BHJ film from the substrate after high-temperature annealing, we characterized the bottom surface (adjacent to ZnO) by optical microscopy (limited resolution) directly through the ITO/glass substrate. The results showed no significant changes, so we just focused on the BHJ/MoO₃ interface before.

After reviewing relevant reports (*Adv. Funct. Mater.* **2013**, *23* (47), 5854-5860; *Nat. Commun.* **2015**, *6* (1), 7269; *Adv. Mater.* **2018**, *30* (48), 1802490; *Nano Lett.* **2021**, *21* (19), 8474-8480.), we found that cross-sectional scanning Kelvin probe microscopy (SKPM) has been employed to characterize the surface potential (SP) difference between functional layers in organic photoelectric devices. The vacuum level alignment within a device can be interpreted by multiplying the SP with electron charge thus revealing the dynamic changes of interfacial band bending. In order to further confirm our proposed mechanism, we have carried out the *operando* cross-sectional SKPM measurements on the Ar⁺ beam milled edge under various operating conditions to visualize the dynamic changes of interfacial band bending of the devices (active layers with a thickness of up to 750 nm are added for clearer results).

The cross-sectional atomic force microscopy (AFM) phase and SKPM images (under AM 1.5G illumination, +2 V) and the extracted SP profiles of as-cast and annealed devices in dark and under illumination at ±2 V are shown below. Compared with the as-cast device, when operating in PM-mode, rapid SP drop/rise at (ZnO/BHJ)/(BHJ/MoO₃) interface in annealed device is observed, indicating sharp upward/downward band bending, and *vice versa* in PV-mode. This can be attributed to the accumulation of photogenerated hole/electron at the interfaces, demonstrating the existence of structural traps (which affect both electrons and holes) at both ZnO/BHJ and BHJ/MoO₃ interfaces in the annealed device.

This result provides a strong support on our proposed mechanism, and we have revised and supplemented the relevant content in manuscript.

Revision:

We have added the *operando* cross-sectional SKPM results to Fig. 2 and Supplementary Fig. 7. We added the detailed discussions in section “Evolution of BHJ films with different annealing durations and the effects on the interfacial band bending” on page 10.

3. The authors should clarify the composition of the aggregates and explain why the aggregates form traps rather than acting as insulating islands.

Response:

Thanks for the comment. We have tried to carry out EDS characterization, hoping to clarify the composition of aggregates through the distribution of characteristic elements (N and F) in Y6. However, this method failed due to the limited detection effect of EDS on light elements.

Therefore, we prepared PBDB-T:Y6 films in different proportions. After annealing at 250°C for 30 min, the surface morphology of the films was observed by optical microscope. As shown below, the amount of aggregates presents on the film surface increases significantly with the increase of Y6 content. This result could be the evidence for proving that aggregates are caused by Y6.

The aggregates affect the contact characteristics of the interface and induce spatial blind alleys (structural traps) at the interfaces. Carriers are hindered (the result is equivalent to being captured) and then accumulate at the interfaces. In fact, in terms of “blind alleys”, it can also be described as “insulating islands”, “dead ends”, *etc.*, and we think those statements are not contradictory. From the function perspective, we unified the term as structural traps referring to previous reports (*Appl. Phys. Lett.* **1998**, 73 (18), 2627-2629).

Revision:

We have added the result to Supplementary Fig. 6.

4. The description of Fig. 4 f-i is missing.

Response: We appreciate the reviewer for the careful review and we have corrected the mistake.

5. PV-type photodiodes feature fast response to light signals. But the markedly increased trap density in the annealed device is supposed to increase the response time. Furthermore, the EQE of the annealed device working in PV-mode is quite low. Therefore, it is recommended that the provided device characteristics be discussed in detail with the reported values to better understand the effect of annealing on device performance.

Response:

This is an important comment. According to the reviewer’s suggestion, we have measured the response time of as-cast OPD (-2 V), as shown below, the t_r/t_f is 4.15/4.19 μ s respectively, which is slightly faster than the dual-mode device (8.64/11.14 μ s, PV-mode). It indicates that the introduced structural traps after annealing do increase the response time (but such a gentle change has little influence on the real applications).

The peak EQE values are 71% and 43% for as-cast and 30-min-annealed devices at -2 V, respectively. The reduced EQE in PV-mode after annealing can be attributed to: 1) the high-temperature annealing process introduces many structural traps at ZnO/BHJ and BHJ/MoO₃ interfaces, limiting carrier transport and collection; 2) bimolecular recombination is intensified in BHJ by poor charge transport.

We have added a detailed discussion to exhibit the advantages and disadvantages of high-temperature annealing strategy comprehensively.

Revision:

We have added the response time result of as-cast OPD to Supplementary Fig. 18 and a detailed discussion on high-temperature annealing strategy in the manuscript.

6. The authors should add more experimental results to demonstrate the “non-close-contact BHJ/MoO₃ interface”.

Response:

This is an important comment. Observing the interface by microscopy directly seems to be the most clear and effective method. So we performed the cross-sectional FIB-TEM characterization of the annealed device, but the results show that there were no visible structural traps at ZnO/BHJ and BHJ/MoO₃ interfaces. We speculate that the size of structural traps is not sufficient to be visually demonstrated by microscopy. Therefore, proving the obstruction and accumulation of carriers at the interfaces directly is a more feasible and rigorous way.

According to *operando* cross-sectional SKPM results, compared with as-cast device, carriers accumulate at both ZnO/BHJ and BHJ/MoO₃ interfaces of dual-mode OPD, which is exactly the result of “non-close-contact”.

Revision:

We have added the *operando* cross-sectional SKPM results and the corresponding discussions in the manuscript.

7. The results indicate long-time annealing caused the decomposition of Y6 which represents the destruction hole transport channels. Accordingly, over-heating may favor hole blocking in PM-mode devices and retard hole transport in PV-mode devices. This trade-off should be discussed. In addition, also, because of this trade-off, the authors are advised to state whether there are ways to further improve the performance of the device.

Response:

Thanks for the reviewer's suggestions. The high-temperature annealing strategy does bring a trade-off between the performance of PM and PV modes. In accordance with Question 5, we have also added a corresponding discussion in revised manuscript.

Based on our proposed device structure of “interfacial structural trap + blocking layer”, subsequent work can focus on quantitative and precise control of the interfacial structural trap through nano-imprinting, laser processing, or other methods to avoid the adverse effects of high-temperature annealing and further optimize the device performance. This is also the direction of our follow-up work.

Reviewer #2:

The purpose of this work is to address the issues of the limited sensitivity of photovoltaic (PV)-type organic photodiodes under dim light and the limited stability of photomultiplication (PM)-type organic photodiodes under strong light. The author proposed an organic photodiode (OPD) that has a PM detection mode when it is operated under a forward bias, and it also acts as a PV-type OPD when it is operated under a reverse bias. The bias-switchable PM- or PV-mode detection in one OPD was realized by introducing thermal-induced interfacial traps and charge blocking layers at the bulk heterojunction (BHJ)/MoO₃ interface. The combined effects lead to the unique PM- and PV-type OPD detection behaviors: e.g., (1) reduced dark current and thereby improved detectivity; (2) enhanced carrier mobility and thus faster response speed of OPD when operating in the PV-mode. (3) high EQE due to the tunneling charge injection in the OPD operated under PM mode (forward bias). The results are interesting. However, the PM-mode OPD shows better photoresponse linearity under illumination of a strong light than that of the PV-mode OPD, while its PV- mode has a better photocurrent linearity under illumination of a weak light than that of its PM-mode, as indicated in Fig 4c. The performance comparisons between single-mode PM-type OPD, single-mode PV-type OPD and the dual-mode OPD are not presented in the manuscript. The aim of the work as highlighted above is not supported fully by the results presented in the present format of the manuscript.

Response:

Thanks for the reviewers' recognition and constructive suggestions. In order to realize bias-switchable PM/PV dual-mode operation, we combine the thermal-induced interfacial structural traps (to achieve PM mode under positive bias) and charge blocking layers (to maintain PV mode under reverse bias). Therefore, the device structure and the name of corresponding single-mode PM-type OPD, single-mode PV-type OPD used in the manuscript are shown in the following table. According to the reviewers' suggestions, we have carried out a series of experiments to quantitatively compare the device performance.

	single-mode PV-type OPD	single-mode PM-type OPD	dual-mode OPD
i.e.	As-cast OPD	MoO ₃ -free OPD	dual-mode OPD
operating bias	-2 V	+2 V	-2 V/+2 V
structural traps	—	√	√
charge blocking layer	√	—	√

The following issues should be addressed before the revision can be considered for a possible publication in Nature Communications:

1. The light power dependence of the photocurrent density measured for the single-mode PM-type OPD and single-mode PV-type OPD should be provided for comparison studies, e.g., compare the results obtained for the dual-mode OPD that is operated under the corresponding PM- and PV- modes. Please discuss the advantages and disadvantages of the dual-mode OPDs compared to that of the single mode OPDs.

Response:

We have carried out $J_{ph}-P_{in}$ measurements of as-cast OPD (single-mode PV-type OPD, -2 V) and MoO₃-free OPD (single-mode PM-type OPD, +2 V), as shown below.

For as-cast OPD, the $J_{ph}-P_{in}$ is linear dependence with linear dynamic range (LDR) of 141dB. The range of dual-mode OPD in PV-mode is comparable with the as-cast OPD, but the $J_{ph}-P_{in}$ is sublinear. This is due to the introduction of blind alleys (structural traps) at the interfaces by high-temperature annealing, which leads to the hindering of photogenerated carrier transport and collection. Thus, the bimolecular recombination in BHJ is aggravated and results in a sublinear $J_{ph}-P_{in}$ dependence (can be processed by data fitting).

The $J_{ph}-P_{in}$ of MoO₃-free OPD is similar to that of dual-mode one in PM-mode, showing a sublinear dependence (119 dB), which is attributed to structural traps at the interfaces. In addition, due to the introduction of the MoO₃ blocking layer in dual-mode OPD, its shallow LUMO level can enhance the accumulation of photogenerated carriers, further improving the PM performance, so the dual-mode OPD has higher photocurrent and wider sublinear range (125 dB).

Revision:

We have added the results to Supplementary Fig. 22, and the detailed discussions on page 20.

2. Please check the figure captions in Fig 4.

Response:

Thank the reviewer very much for the careful review. We have checked and corrected the figure caption in Fig. 4.

3. What is the mechanism accounting for the reduced dark current in the annealed OPDs operating in either PM or PV mode?

Response:

This is an important comment. The as-cast OPD exhibits a smooth and flat BHJ surface and has close contact with ZnO and MoO₃ layers, thus the carriers can easily pass through the ZnO/BHJ and BHJ/MoO₃ interfaces without hindrance. On the contrary, the annealed OPD forms numerous structural traps at the interfaces, blocking the charge injection from external circuit. (According to the supplementary *operando* cross-sectional SKPM results, structural traps also exist at the ZnO/BHJ interface, see Question 8 for details.).

In PM-mode (forward bias), the electrons/holes injected from the ITO/Ag electrode are blocked at the ZnO/BHJ and BHJ/MoO₃ interfaces, respectively, thereby reducing the J_d (Figure a below).

In PV-mode (reverse bias), the holes/electrons injected from ITO/Ag electrode are mainly blocked by ZnO and MoO₃ interface layers, so J_d is lower than in PM-mode. In addition, J_d is further reduced compared to as-cast device due to the synergistic effect of structural traps at the ZnO/BHJ and BHJ/MoO₃ interfaces (Figure b below).

Revision:

We have revised the relevant descriptions according to the *operando* cross-sectional SKPM results.

4. The appearance of the microscopic aggregation is usually associated with the degradation in OPDs, which is observed in the thermal annealed OPDs as presented in the manuscript. The author shall discuss the durability of the dual-mode OPDs as compared to that of the as-casted devices.

Response:

As suggested by the reviewer, we have conducted the durability measurement of the devices, and the results are shown below ($N = 5$). The as-cast and dual-mode OPDs (150 nm-thick, 30 min-annealed) were placed in air environment at room temperature for two months, and the device performance had different degrees of attenuation.

In contrast, the attenuation degree of EQE and D^* of dual-mode OPDs in both PM and PV modes is slightly higher than that of as-cast OPDs. This can be attributed to the fact that high-temperature annealing intensifies the degradation of organic materials, especially Y6, in the active layer, thus adversely affecting the durability of the device.

Revision:

We have added the durability result to Supplementary Fig. 12 and the corresponding discussion on page 14.

5. What is the meaning of “the aggregates that continuously grow beyond the film thickness” in line 185-186, on page 9?

Response:

With the increase of annealing time, the size of the aggregates increases continuously. After 120 min, the aggregates can even grow to a particle size larger than the thickness of active layer, and embed throughout the entire active layer, which can also be observed from the bottom side (through the ITO glass). Similar growth process has previously been reported (*Mater. Horiz.*, **2021**, 8, 1272–1285).

Revision:

We have revised the corresponding description on page 10: “After 120 min, the aggregates can even grow to a particle size larger than the thickness of active layer, and embed throughout the entire active layer, which can also be observed from the bottom side (through the ITO glass, Supplementary Fig. 4)”.

and page 13: “The decrease in R_{ct} with prolonged annealing process (over 30 min) can be attributed to the continuous growth of aggregates. The large-sized aggregates embed throughout the entire active layer, forming short circuit paths⁴¹”.

6. A reverse bias of -40 V was applied to the OPD, as indicated in supplementary Fig 2c. Please explain why such a high voltage is required for the operation of the OPDs?

Response:

This is an important comment. The OPD in supplementary Fig. 2c is MoO₃-free device with structure of ITO/ZnO/BHJ(530 nm)/Ag. We designed this set of experiments to qualitatively verify the importance of MoO₃ blocking layer for maintaining PV-mode under reverse bias, but the thickness of active layer was not strictly controlled. Due to the thick active layer, the photocurrent of the device was suppressed under reverse bias, thus a high bias was required to induce band bending and charge tunneling injection (-40 V in this case).

As supplement, we have prepared MoO₃-free OPD with 150 nm-thick active layer, as shown below. In this case, the MoO₃-free OPD cannot maintain PV-mode at a low bias of only -2 V.

Furthermore, we can comprehensively summarize the functions of MoO₃ blocking layer, as follows:

a) Block the charge injection from external circuit in dark to reduce the dark current;

Experimental evidences: the J_d values of MoO₃-free OPD increase sharply under reverse bias.

- b) Block the charge tunneling from external circuit under illumination to maintain PV-mode under reverse bias;

Experimental evidences: the MoO₃-free OPD cannot maintain PV-mode at just -2 V, as the EQE values exceeds 100% in the range of 360-400 nm. When the bias is increased to -5 V, the EQE values in the whole measured range are almost the same as the PM-mode of +2 V.

- c) Further enhance the accumulation of photogenerated holes at BHJ/MoO₃ interface under forward bias due to its shallow LUMO, and thus improve PM effect.

Experimental evidences: compared with the PM-mode (+2 V) of dual-mode OPD, the EQE values of MoO₃-free OPD are much reduced.

Revision:

We have added the performance of 150-nm-thick MoO₃-free OPD to Supplementary Fig. 2b, and the corresponding discussions on page 8.

“As shown in Supplementary Fig. 2, the MoO₃-free OPDs exhibit PM effect under both forward and reverse biases. Thus, the blocking layers play a vital role in maintaining PV response characteristic under reverse bias and ensuring the bias-switchability of operation mode. Furthermore, taking MoO₃ for example, the functions of blocking layer can be comprehensively summarized: 1) block the charge injection from external circuit in dark to reduce the dark current; 2) block the charge tunneling from external circuit under illumination to maintain PV-mode under reverse bias; 3) further enhance the accumulation of photogenerated holes at BHJ/MoO₃ interface under forward bias due to its shallow lowest unoccupied molecular orbital (LUMO), and thus improve the PM effect.”

7. What are the -3dB cut-efficiencies measured for the dual-mode OPD operated under its PM- and PV-modes? Are the -3dB cut-efficiencies measured for the dual-mode OPD operated under its PM- or PV-mode similar to that obtained for the corresponding discrete PM and PV OPDs?

Response:

Thanks for the comment. We have measured the -3dB cutoff frequencies of the OPDs under 850 nm light. The results show that the -3dB cutoff frequencies of dual-mode OPD operating in PM and PV modes are similar to that of the corresponding discrete PM and PV OPDs, respectively.

The as-cast OPD (single-mode PV-type OPD, -2 V) and dual-mode OPD operating in PV-mode (-2 V) both exhibit similar -3dB cutoff frequencies of exceeding 100 kHz (153.1/113.5 kHz) due to the fast response speed, and the as-cast one has slightly larger frequency due to the faster response speed.

Owing to the time-consuming carrier accumulation and band tunneling processes in PM effect, the -3dB cutoff frequencies of dual-mode OPD operating in PM-mode (+2 V) and MoO₃-free OPD (single-mode PM-type OPD, +2 V) both decrease significantly to 8.9 kHz and 20.8 kHz respectively. The shallow LUMO level of MoO₃ allows dual-mode OPD to accumulate more carriers at the interface and thus obtain higher EQE, but this also requires a longer accumulation time, and therefore has a lower -3dB cutoff frequency than MoO₃-free OPD.

Revision:

We have added the result of -3dB cutoff frequencies to Supplementary Fig. 20, and the corresponding discussions on page 19.

8. Did the thermal-induced interfacial traps occur only at the BHJ/MoO₃ interface or the traps also formed at the ZnO/BHJ interface in the dual-mode OPD? If the traps exist at both BHJ/MoO₃ and ZnO/BHJ interfaces, which interface plays a dominant role in determining the PM behavior in the dual-mode OPD?

Response:

We are grateful for the reviewer's comment.

In the previous manuscript, we have carried out a series of characterization on the top surface (adjacent to MoO₃) of the active layer (including optical microscopy, SEM

and laser scanning confocal microscope), and obtained obvious morphological changes. However, since it is very difficult to peel off the BHJ film from the substrate after high-temperature annealing, we characterized the bottom surface (adjacent to ZnO) by optical microscopy (limited resolution) directly through the ITO/glass substrate. The results showed no significant changes, so we just focused on the BHJ/MoO₃ interface before.

After reviewing relevant reports (*Adv. Funct. Mater.* **2013**, 23 (47), 5854-5860; *Nat. Commun.* **2015**, 6 (1), 7269; *Adv. Mater.* **2018**, 30 (48), 1802490; *Nano Lett.* **2021**, 21 (19), 8474-8480.), we found that cross-sectional scanning Kelvin probe microscopy (SKPM) has been employed to characterize the surface potential (SP) difference between functional layers in organic photoelectric devices. The vacuum level alignment within a device can be interpreted by multiplying the SP with electron charge thus revealing the dynamic changes of interfacial band bending. In order to further confirm our proposed mechanism, we have carried out the *operando* cross-sectional SKPM measurements on the Ar⁺ beam milled edge under various operating conditions to visualize the dynamic changes of interfacial band bending of the devices (active layers with a thickness of up to 750 nm are added for clearer results).

The cross-sectional atomic force microscopy (AFM) phase and SKPM images (under AM 1.5G illumination, +2 V) and the extracted SP profiles of as-cast and annealed devices in dark and under illumination at ±2 V are shown below. Compared with the as-cast device, when operating in PM-mode, rapid SP drop/rise at (ZnO/BHJ)/(BHJ/MoO₃) interface in annealed device is observed, indicating sharp upward/downward band bending, and *vice versa* in PV-mode. This can be attributed to the accumulation of photogenerated hole/electron at the interfaces, demonstrating the existence of structural traps (which affect both electrons and holes) at both ZnO/BHJ and BHJ/MoO₃ interfaces in the annealed device.

It can be observed that the energy level varies more obviously at ZnO/BHJ interface, indicating greater charge accumulation. Therefore, we speculate that the ZnO/BHJ interface dominates.

Revision:

We have added the *operando* cross-sectional SKPM results to Fig. 2 and Supplementary Fig. 7. We added the detailed discussions in section “Evolution of BHJ films with different annealing durations and the effects on the interfacial band bending” on page 10.

Reviewer #3:

This is the reviewer report for then manuscript entitled “Organic Photodiodes with Bias-Switchable Photomultiplication and Photovoltaic Modes” for Nature Communications.

In brief, the authors of this paper advance organic photodiodes based on Bulk Heterojunction (BHJ) devices which operate in a photovoltaic (PV) mode or a photomultiplication (PM) mode depending on the polarity of the applied voltage. Specifically, by applying a forward bias the devices show a PM effect resulting in very large EQEs, while the “normal” PV mode is obtained when device is operated in the reverse bias. This behaviour is obtained by conducting an annealing treatment of the BHJ at a relatively high temperature, and is attributed to the formation of thermal-induced traps at the BHJ/MoO₃ interface.

While the idea and concept are both very interesting and the experimental results looks promising, my main criticism is that the proposed working principle and physical mechanism behind the PM effect in forward bias seems lacking and not supported by convincing experimental evidence.

Response:

Thanks for the reviewer’s commendatory comments and the suggestions. We have supplemented the relevant characterization and experiments to confirm our proposed working mechanism, as detailed in the following point-by-point response.

Detailed comments:

1. Traps are usually either acceptor-type or donor-type, meaning that they will either be electron traps or hole traps, respectively. An acceptor-type (electron) trap will be neutral if empty and negatively charged when occupied by an electron, whereas a donor-type trap is positively charged when empty (*i.e.*, occupied by a hole) and neutral when occupied by an electron. In this work, however, it is suggested that the traps can act as both electron and hole traps, simply based on the applied bias. How can this be the case? Such claims require more theoretical and experimental justification.

Response:

We appreciate the concern that the reviewer raised.

The definition of our proposed “interfacial traps” are actually structural traps, which are different with those reported by many others. Most of the reported PM-type OPDs are based on the extreme unbalanced ratio of donor and acceptor, and the acceptor-type (electron) traps or donor-type (hole) traps are introduced to achieve PM effect. That is the trap pointed by the reviewer.

In this work, the thermal-induced “interfacial traps” are actually “structural traps”. The surface morphology of BHJ active layer was tuned after high-temperature annealing, affecting the contact characteristics of the interface and thus inducing spatial blind alleys (structural traps) at the interfaces. The carriers (either electrons or holes) will be hindered and then accumulate at the interfaces. The process can be described as, “carriers are captured by the structural traps and then accumulation at the interface”.

As mentioned above, the most prominent advantage of this “structural traps” is that they have the same trapping effect on electrons and holes, and can tune the trapped carrier type by bias direction. It is confirmed that structural traps exist at both ZnO/BHJ and BHJ/MoO₃ interfaces by the supplementary *operando* cross-sectional scanning Kelvin probe microscopy (SKPM, see Question 2 for details). The working mechanism of the traps in the two interfaces is similar, and the BHJ/MoO₃ interface is discussed below as example.

When in the dark, whether holes (forward bias, PM-mode) or electrons (reverse bias, PV-mode) transported from external circuit to BHJ/MoO₃ interface can be captured by structural traps (blind alleys), thus reducing the dark current of the device at both forward and reverse bias.

When under illumination, whether photogenerated electrons (forward bias, PM-mode) or photogenerated holes (reverse bias, PV-mode) can also be captured by

structural traps at the BHJ/MoO₃ interface. The results are as follows: 1) the collection efficiency of photogenerated carriers in PV mode is affected, and the EQE values are reduced compared with as-cast device; 2) Under forward bias (PM mode), photogenerated carriers are hindered and then accumulate at the interface, causing band bending and carrier tunneling from the external circuit to obtain PM effect.

We realize that the discussion on this part was not clearly enough. Therefore, in the revised manuscript, we have supplemented relevant characterization and description.

Revision:

We have supplemented the relevant characterization and description.

2. Why are thermal-induced traps only formed at the BHJ/MoO₃ interface?

Response:

We appreciate for the reviewer' comment.

In the previous manuscript, we have carried out a series of characterization on the top surface (adjacent to MoO₃) of the active layer (including optical microscopy, SEM and laser scanning confocal microscope), and obtained obvious morphological changes. However, since it is very difficult to peel off the BHJ film from the substrate after high-temperature annealing, we characterized the bottom surface (adjacent to ZnO) by optical microscopy (limited resolution) directly through the ITO/glass substrate. The results showed no significant changes, so we just focused on the BHJ/MoO₃ interface before.

After reviewing relevant reports (*Adv. Funct. Mater.* **2013**, *23* (47), 5854-5860; *Nat. Commun.* **2015**, *6* (1), 7269; *Adv. Mater.* **2018**, *30* (48), 1802490; *Nano Lett.* **2021**, *21* (19), 8474-8480.), we found that cross-sectional scanning Kelvin probe microscopy (SKPM) has been employed to characterize the surface potential (SP) difference between functional layers in organic photoelectric devices. The vacuum level alignment within a device can be interpreted by multiplying the SP with electron charge thus revealing the dynamic changes of interfacial band bending. In order to further confirm our proposed mechanism, we have carried out the *operando* cross-sectional SKPM measurements on the Ar⁺ beam milled edge under various operating conditions to visualize the dynamic changes of interfacial band bending of the devices (active layers with a thickness of up to 750 nm are added for clearer results).

The cross-sectional atomic force microscopy (AFM) phase and SKPM images (under AM 1.5G illumination, +2 V) and the extracted SP profiles of as-cast and

annealed devices in dark and under illumination at ± 2 V are shown below. Compared with the as-cast device, when operating in PM-mode, rapid SP drop/rise at (ZnO/BHJ)/(BHJ/MoO₃) interface in annealed device is observed, indicating sharp upward/downward band bending, and *vice versa* in PV-mode. This can be attributed to the accumulation of photogenerated hole/electron at the interfaces, demonstrating the existence of structural traps (which affect both electrons and holes) at both ZnO/BHJ and BHJ/MoO₃ interfaces in the annealed device.

This result provides a strong support on our proposed mechanism, and we have revised and supplemented the relevant content in manuscript.

Revision:

We have added the *operando* cross-sectional SKPM results to Fig. 2 and Supplementary Fig. 7. We added the detailed discussions in section “Evolution of BHJ films with different annealing durations and the effects on the interfacial band bending” on page 10.

3. The authors use SCLC to estimate the trap density in both as-cast and annealed devices (Fig 3f). However, Eq. (2) assume that the traps are distributed throughout the bulk of the active layer, rather than interfacial traps.

Response:

We are very grateful for the reviewer’s reminding and correction. After checking the relevant literature (*Physical Review* **1956**, *103* (6), 1648-1656; *Adv. Mater.* **2021**, *33* (14), 2008134) and obtaining a deeper understanding of the trap density measurement by SCLC, we agree that the SCLC method is not suitable for this work. Therefore, we removed it from the manuscript.

4. It is not clear why the J_{ph} has a sublinear dependence with P_{in} in the relevant regime. Since trap-assisted recombination is usually considered a (nearly) first-order recombination process, I would have expected the J_{ph} vs P_{in} to display a (almost) linear dependence. This is not seen for either PV or PM mode. In fact, the J_{ph} vs P_{in} curves in the PV mode (in Fig 4c) seem to show close to a slope of 1/2 on the log-log scale, which is what one would be expected if the current is strongly limited by bimolecular recombination. This would suggest that there is something wrong/missing in the working mechanism proposed by the authors?

Response:

Thanks for the comment. As our answer to reviewer's Question 1, different from other reported PM-type OPDs, the traps we introduce at the interfaces are structural traps (blind alleys), and the key working mechanism is to hinder the carrier transport. Thus the bimolecular recombination will gain favorable competition against the extraction and collection of photogenerated carriers, resulting in a sublinear J_{ph} - P_{in} dependence. (*Adv. Mater.* **2022**, 34 (13), 2109516; *Adv. Mater.* **2017**, 29 (46), 1704051.)

As a supplement, we have carried out J_{ph} - P_{in} measurement of as-cast device (-2 V), as shown below. In the absence of structural traps, the J_{ph} - P_{in} of the device presents a linear dependence, and this can further validate our proposed working mechanism.

Revision:

We have added the J_{ph} - P_{in} dependence result of as-cast device to Supplementary Fig. 23.

5. Based on GIWAXS, the authors conclude that there is a higher regularity and crystallinity of annealed films (beneficial for improving the mobility). These findings seem contradictory to the other experimental data that i) the EQE in PV mode is lower

compared to as-cast device, and ii) the suggestion that there is a large number of traps and worse surface morphology in annealed devices. Also, the sublinear J_{ph} vs P_{in} curve is consistent with poor charge transport, suggesting a low mobility in the annealed devices.

Response:

This is an important comment. The GIWAXS only characterized the properties of BHJ active layer, while the contradictions proposed by the reviewer are caused by the ZnO/BHJ and BHJ/MoO₃ interfaces. The details are as follows:

1) the EQE in PV mode is lower compared to as-cast device:

Although high-temperature annealing improves the regularity and crystallinity of BHJ bulk, which is beneficial for improving the mobility, it also introduces blind alleys (structural traps) at the interfaces. Thus, the carrier transport and collection are hindered at interfaces, and the bimolecular recombination is aggravated in BHJ. As a result, the EQE in PV-mode is reduced.

2) the suggestion that there is a large number of traps and worse surface morphology in annealed devices:

The annealed devices do have worse surface morphology, but the structural traps are introduced at the interfaces rather than inside the BHJ. We carried out TRPL to characterize the BHJ films, as shown below. The 30-min annealed film shows reduced average lifetime τ_{avg} , which indicates decreased trap density in the BHJ bulk and thus weaker time-consuming trap-assisted recombination, consistent with the GIWAXS characterization.

3) the sublinear J_{ph} vs P_{in} curve is consistent with poor charge transport, suggesting a low mobility in the annealed devices:

The annealed devices do have poor charge transport, but it occurs mainly at the ZnO/BHJ and BHJ/MoO₃ interfaces, resulting in a low mobility in the annealed devices.

Revision:

We have supplemented the TRPL result to Supplementary Fig. 17, and the corresponding discussions on page 17.

“In addition, the time-resolved photoluminescence (TRPL) spectra (Supplementary Fig. 18) show reduced average lifetime τ_{avg} of 30-min-annealed BHJ film, which indicates decreased defect density in the BHJ bulk and thus suppressed time-consuming trap-assisted recombination, consistent with the GIWAXS results⁵².”

6. Finally, I also got confused about the charge transfer resistance R_{ct} that the authors extracted from impedance spectroscopy. The increased R_{ct} is attributed to restricted injection in annealed devices. However, the R_{ct} in Fig 2d is approximately only increased by a factor of 3 compared to as-cast case. I fail to see how such a relatively small change can give rise to reduction of J_d in the forward-bias by several orders of magnitude?

Response:

This is an inspiring comment. The previous EIS results were measured in dark with no bias applied. After reviewing the relevant literature, we realized that a certain bias should be applied to the device during testing. We have re-conducted the experiment at +2 V, as shown below. The R_{ct} values are markedly enhanced after annealing and the differences between R_{ct} values of devices with different annealing durations are more obvious, corresponding to the change of J_d .

Revision:

We have amended the EIS result to Supplementary Fig. 9.

REVIEWER COMMENTS

Reviewer #1 (Remarks to the Author):

The revised manuscript has added more supplemental information to support their idea and reply to the previous comments. However, there are still several issues that should be carefully addressed before this paper can be recommended for publication. Particularly, some responses to the previous comments are only listed in the authors' Rebuttal, and there is lack of the corresponding revisions in the manuscript. The other comments are shown below.

1. The authors explained the advantages of the designed dual-mode photodiodes in the Response Letter. However, the related issues should be added in the context of the revised manuscript.
2. Ref. 37 is missing, which should be added.
3. In the paper, the dual-mode bias-switchable OPD was operated in PM-mode under forward bias and in the PV-mode under reverse bias. For the practical application, the photodetectors are commonly operated under the same DC bias. How to be compatible with the conventional applications for the present OPDs with the selection of a suitable mode?
4. Fig. 1b shows the schematic diagram of the working mechanisms of the dual-mode OPDs in PV and PM modes. However, it is not correct for the PM mode in the present figure. The authors should correct the errors.
5. The authors mentioned the introduction of structural traps at ZnO/BHJ and BHJ/MoO₃ interfaces after high-temperature annealing treatment. What is the physical meaning or origin of these structural traps induced by annealing treatment, although the authors attributed them as spatial blind alleys? In fact, the reviewer in the previous comments had asked the similar question, while the authors' rebuttal is not very clear.
6. The PM-mode performance for as-cast device should be added in Figure 1d and 1c as a reference.
7. One important issue is that the dual-mode performance is due to the introduction of interfacial traps for the charge accumulation and thus tunneling. However, the response performance of the devices under the continuous pulse operation should be tested to clarify the influence of the trap charges for the detection capability.

Reviewer #2 (Remarks to the Author):

Authors have addressed the questions raised for the previous version, and made substantial changes following the reviewers' comments. The technical quality of the revision is sound. In my opinion, the revised manuscript meets the quality requirement for publication in Nature Communications.

Reviewer #3 (Remarks to the Author):

The authors have addressed my comments and clarified and supplemented their proposed working mechanism with additional experimental findings. One final comment is that the authors should consider replacing the word "time-consuming" with something more appropriate when referring to enhanced recombination. Otherwise, I support publication of this paper.

Response to reviewers

TITLE: Organic Photodiodes with Bias-Switchable Photomultiplication and Photovoltaic Modes

(Nature Communications manuscript NCOMMS-22-53601A-Z)

Dear reviewers,

Thank you all very much for your time and valuable suggestions for improving the quality of our manuscript submitted to your esteemed peer reviewed journal. The list of point-by-point response to the reviewers' comments is provided below. All corrections/modifications made in the revised manuscript were highlighted with **Yellow Color**.

Reviewers' comments:

Reviewer #1:

The revised manuscript has added more supplemental information to support their idea and reply to the previous comments. However, there are still several issues that should be carefully addressed before this paper can be recommended for publication. Particularly, some responses to the previous comments are only listed in the authors' Rebuttal, and there is lack of the corresponding revisions in the manuscript. The other comments are shown below.

Response:

Thanks for the reviewer's commendatory comments and suggestions. We have revised the manuscript according to the previous and current comments, as detailed in the following point-by-point response.

1. The authors explained the advantages of the designed dual-mode photodiodes in the Response Letter. However, the related issues should be added in the context of the revised manuscript.

Response and revision:

Thanks for the reviewer's reminding. We have added the corresponding description on page 21. we added the following sentence after the sentence "...keep the photocurrent stable at a relatively balanced level."

"For the dim scenarios like night vision, the PM-mode with high EQE can avoid the use of amplifying circuits; for the strong-light scenarios like under direct sunlight, the low-power-consumption PV-mode can avoid potential damages (heat-dissipation,

breakdown, etc.) to the organic active layer. This is beneficial for reducing costs and the burden of signal processing.”

2. Ref. 37 is missing, which should be added.

Response:

Thanks for the reviewer's attention. Due to the rules of double-blind review, we need to hide Ref. 37 in the manuscript, which is related to the authors' information, and attach it in the cover letter to the editor.

3. In the paper, the dual-mode bias-switchable OPD was operated in PM-mode under forward bias and in the PV-mode under reverse bias. For the practical application, the photodetectors are commonly operated under the same DC bias. How to be compatible with the conventional applications for the present OPDs with the selection of a suitable mode?

Response:

In practical application, the selection of operating mode can be realized by an adjustable power supply (including adjustable bias direction and value), which can be achieved by integrated circuits.

Take the circuit we designed as an example (shown below), the input is a DC bias, the direction of the output bias (the bias of OPD) can be changed through the switch SW1, and the output bias value can be adjusted by the sliding rheostat RP1 and RP2.

Similarly, the input bias can also be DC/AC, and the adjustment mode of bias direction and value can also be knob, numerical control, etc., enough to meet various needs. Therefore, the compatibility of the OPD in application can be well satisfied.

4. Fig. 1b shows the schematic diagram of the working mechanisms of the dual-mode OPDs in PV and PM modes. However, it is not correct for the PM mode in the present figure. The authors should correct the errors.

Response:

Thanks to the reviewer's careful review, we have corrected the errors in the revised manuscript.

5. The authors mentioned the introduction of structural traps at ZnO/BHJ and BHJ/MoO₃ interfaces after high-temperature annealing treatment. What is the physical meaning or origin of these structural traps induced by annealing treatment, although the authors attributed them as spatial blind alleys? In fact, the reviewer in the previous comments had asked the similar question, while the authors' rebuttal is not very clear.

Response:

We are sorry for the unclear explanations.

The as-cast BHJ film exhibits a smooth and flat surface and close contact with ZnO and MoO₃ layers, thus the carriers can easily pass through the ZnO/BHJ and BHJ/MoO₃ interfaces without hindrance. After high-temperature annealing treatment, the surface morphology of BHJ active layer is tuned to be rather rough, affecting the contact characteristics of the interface and thus inducing numerous spatial blind alleys (structural traps) at the ZnO/BHJ and BHJ/MoO₃ interfaces. The structural traps have the same trapping effect on electrons and holes, and can tune the trapped carrier type by bias direction.

The existence of structural traps at both ZnO/BHJ and BHJ/MoO₃ interfaces has been proved by the *operando* cross-sectional scanning Kelvin probe microscopy. Since the working mechanism at the two interfaces is similar, the BHJ/MoO₃ interface is discussed below as example.

When in the dark, whether holes (forward bias, PM-mode) or electrons (reverse bias, PV-mode) transported from external circuit to BHJ/MoO₃ interface can be captured by structural traps (spatial blind alleys), thus reducing the dark current of the device at both forward and reverse bias.

When under illumination, whether photogenerated electrons (forward bias, PM-mode) or photogenerated holes (reverse bias, PV-mode) can also be captured by structural traps at the BHJ/MoO₃ interface. Therefore: 1) under reverse bias (PV mode), the photogenerated electrons/holes in BHJ film are extracted by the ITO/Ag electrode to generate photocurrent. while the collection efficiency of photogenerated carriers is affected by structural traps, thus the EQE values are reduced compared with as-cast

device; 2) under forward bias (PM mode), photogenerated carriers are hindered and then accumulate at the interface, causing band bending and carrier tunneling from the external circuit to obtain PM effect.

Revision: We have modified the corresponding description on page 6.

6. The PM-mode performance for as-cast device should be added in Figure 1d and 1c as a reference.

Response:

Due to the high dark current density of the as-cast device under forward bias, the photocurrent signal cannot be effectively distinguished and acquired (Figure 1c). Thus, the as-cast device can only be operated in PV-mode under reverse bias, and there is no PM-mode performance under forward bias.

7. One important issue is that the dual-mode performance is due to the introduction of interfacial traps for the charge accumulation and thus tunneling. However, the response performance of the devices under the continuous pulse operation should be tested to clarify the influence of the trap charges for the detection capability.

Response:

Thanks for the comment.

Compared with PV-mode, the relatively time-consuming charge accumulation and tunneling process in PM-mode will affect the response speed of device. We have supplemented the response performance of the typical dual-mode device under continuous pulse signal (850 nm) in the modulation frequency range of 1 Hz-1 kHz as shown below.

When operating in PV-mode, the output of device can follow the on-off switching of the incident signal and achieve steady state dark current and photocurrent even at the frequency of 1 kHz. While when operating in PM-mode, the response begins to

decrease at 1 kHz and cannot fully reach/decay to the original photocurrent/dark current due to the reduced response speed. We have amended the performance in the revised manuscript.

Furthermore, the response characteristics of the device under the continuous pulse signal can also be demonstrated by the -3dB cutoff frequency, defined as the frequency at which the output of device is attenuated to 0.707 of the original amplitude, (Supplementary Figure 21, shown below). The corresponding descriptions in the manuscript are as follows

“Both the as-cast OPD and dual-mode OPD (PV-mode) exhibit similar -3dB cutoff frequencies of exceeding 100 kHz (153.1/113.5 kHz), and the as-cast one has slightly larger frequency due to the faster response speed. Owing to the time-consuming carrier accumulation and band tunneling processes in PM effect, the -3dB cutoff frequencies of dual-mode OPD operating in PM-mode and 30-min-annealed MoO₃-free OPD (+2 V) both decrease significantly to 8.9 kHz and 20.8 kHz respectively. The shallow LUMO level of MoO₃ allows dual-mode OPD to accumulate more carriers at the interface and thus obtain higher EQE, but this also requires a longer accumulation time, and therefore has a lower -3dB cutoff frequency than MoO₃-free OPD.”

Revision:

We have added the response performance to Supplementary Fig. 20 and the corresponding discussion on page 19.

Reviewer #2

Authors have addressed the questions raised for the previous version, and made substantial changes following the reviewers' comments. The technical quality of the revision is sound. In my opinion, the revised manuscript meets the quality requirement for publication in Nature Communications.

Response:

We appreciate the reviewer's commendatory comments and the recommendation of publication.

Reviewer #3

The authors have addressed my comments and clarified and supplemented their proposed working mechanism with additional experimental findings. One final comment is that the authors should consider replacing the word "time-consuming" with something more appropriate when referring to enhanced recombination. Otherwise, I support publication of this paper.

Response and revision:

Thanks for the reviewer's constructive suggestion and the recommendation of publication. We have replaced the word "time-consuming" to "the relatively slower" following the reviewer's suggestion.